# Post-ischemic ubiquitination at the postsynaptic density reversibly influences the activity of ischemia-relevant kinases
Luvna Dhawka [1,5], Victoria Palfini[1,5], Emma Hambright [1], Ismary Blanco[1], Carrie Poon[1], Anja Kahl[1], Ulrike Resch[2], Ruchika Bhawal[3], Corinne Benakis[1,4], Vaishali Balachandran[1], Alana Holder[1], Sheng Zhang [3], Costantino Iadecola [1] & Karin Hochrainer [1] ✉

Ubiquitin modifications alter protein function and stability, thereby regulating cell homeostasis and viability, particularly under stress. Ischemic stroke induces protein ubiquitination at the ischemic periphery, wherein cells remain viable, however the identity of ubiquitinated proteins is unknown. Here, we employed a proteomics approach to identify these proteins in mice undergoing ischemic stroke. The data are available in a searchable web interface (https://hochrainerlab.shinyapps.io/StrokeUbiOmics/). We detected increased ubiquitination of 198 proteins, many of which localize to the postsynaptic density (PSD) of glutamatergic neurons. Among these were proteins essential for maintaining PSD architecture, such as PSD95, as well as NMDA and AMPA receptor subunits. The largest enzymatic group at the PSD with elevated post-ischemic ubiquitination were kinases, such as CaMKII, PKC, Cdk5, and Pyk2, whose aberrant activities are well-known to contribute to post-ischemic neuronal death. Concurrent phospho-proteomics revealed altered PSD-associated phosphorylation patterns, indicative of modified kinase activities following stroke. PSD-located CaMKII, PKC, and Cdk5 activities were decreased while Pyk2 activity was increased after stroke. Removal of ubiquitin restored kinase activities to pre-stroke levels, identifying ubiquitination as the responsible molecular mechanism for post-ischemic kinase regulation. These findings unveil a previously unrecognized role of ubiquitination in the regulation of essential kinases involved in ischemic injury.

Stroke is a leading cause of death and long-term disability in the US and worldwide[1]. Most strokes are ischemic, resulting from cerebral artery occlusion[2]. Current FDA-approved treatments effective in ameliorating ischemic stroke outcomes focus on early re-establishment of blood flow (reperfusion) by intravenous thrombolysis and endovascular thrombectomy[3,4]. However, due to a narrow therapeutic time window, concerns about hemorrhagic transformation, and, in the case of thrombectomy, the need for specialized equipment and expertise, these treatments are not available to most stroke patients[2,5,6]. There is currently no effective therapy to increase post-ischemic neuronal viability and function[7,8]. Hence, there is a need for additional treatment options, particularly those that protect brain cells from the damaging consequences of ischemia[9].

Ischemic stroke leads to a rapid and sustained shutdown of protein synthesis in the periphery of the ischemic territory[10–12], the brain area that is functionally impaired by stroke but considered viable and salvageable[13]. With no opportunity to synthesize new proteins, cells located in this region use posttranslational modifiers to adapt protein function in an attempt to maintain homeostasis. One such modifier is ubiquitin, the conjugation of which to target proteins is increased in potentially viable regions surrounding the ischemic core[14,15]. Ubiquitination of proteins is involved in many essential cellular processes, such as signal transduction, protein trafficking, DNA damage response, protein degradation, and kinase activity regulation, particularly under stress[16–22]. However, the role of ubiquitination in ischemic stroke is poorly understood[16]. While we recently reported that

[1]Feil Family Brain and Mind Research Institute, Weill Cornell Medicine, New York, NY, USA. [2]Center for Physiology and Pharmacology, Medical University of Vienna, Vienna, Austria. [3]Institute of Biotechnology, Cornell University, Ithaca, NY, USA. [4]Present address: Institute for Stroke and Dementia Research, Ludwig-Maximilians-University Munich, Munich, Germany. [5]These authors contributed equally: Luvna Dhawka, Victoria Palfini. ✉e-mail: kah2015@med.cornell.edu

the main mechanism underlying enhanced post-ischemic ubiquitination is inhibition of deubiquitinases[23], the identity of the target proteins and the impact of ubiquitination on their function remains to be elucidated.

Here, we identified the post-ischemic ubiquitinated proteome in a transient focal ischemia mouse model, using ubiquitin enrichment combined with nano-liquid chromatography (LC)-mass spectrometry (MS)/MS and bioinformatics analyses. We found that ischemia increases the ubiquitination of proteins that are predominantly located at the PSD, many being kinases well known to participate in ischemic injury. We further show that ischemia not only alters the phospho-proteome concurrently with ubiquitination at the PSD but also directly regulates the activity of ischemia-relevant kinases through ubiquitination. These findings support the role of post-ischemic ubiquitination in kinase regulation at the PSD, which may critically influence post-ischemic neuronal function and survival.

## Results
### Ischemic stroke increases the ubiquitination of neuronal proteins
To identify proteins with altered ubiquitination after ischemia, we subjected mice to either sham surgery or focal cerebral ischemia by middle cerebral artery occlusion (MCAO) followed by 1 h reperfusion, the time point when post-ischemic ubiquitination is well established[14,15]. We then extracted neocortical detergent-resistant fractions in sham-operated mice and in mice with MCAO ipsilateral and contralateral to the stroke, and processed the samples for the enrichment of ubiquitin-remnant peptides

by diglycyl-lysine (K-ε-GG) antibody precipitation[24] and subsequent nanoLC–MS/MS analysis (Fig. 1a). We chose this fraction based on previous studies that showed prominent ubiquitination of detergent-insoluble proteins at 1 h reperfusion[14,15], confirmed here in samples used in our proteomic analysis (Fig. 1b). Across 3 MS runs, peptides obtained from Triton-resistant samples not enriched for ubiquitin were similarly abundant in sham, ipsi- and contralateral cortices (Fig. 1c; global peptides listed in Supplementary Data 1, tab 1), showing that the overall abundance of detergent-insoluble proteins after ischemia is not changed. In contrast, and in line with Western Blot data (Fig. 1b and[14,25]), we found a higher number of ubiquitinated peptides ipsilateral to the stroke compared to sham and contralateral cortices by MS (Fig. 1c; ubiquitinated peptides listed in Supplementary Data 1, tab 2). Unsupervised hierarchical clustering between ubiquitinated peptide abundances resulted in a heatmap in which sham and contralateral datasets formed a separate cluster from ipsilateral to the stroke samples (Fig. 1d), confirming the altered ubiquitination pattern in the ischemic cortex. This pattern was preserved at the protein abundance level as well (Fig. 1e). Proteins with changed ubiquitination after ischemic stroke, including specific ubiquitination sites, can be interrogated through a user-friendly search database with the URL https://hochrainerlab.shinyapps.io/StrokeUbiOmics/. Next, we asked how many proteins exhibit changed ubiquitination levels in our datasets. We found a total of 369 high-confidence ubiquitinated proteins that appeared in all three MS runs (listed in Supplementary Data 1, tab 3). Out of these, only 3 ubiquitinated

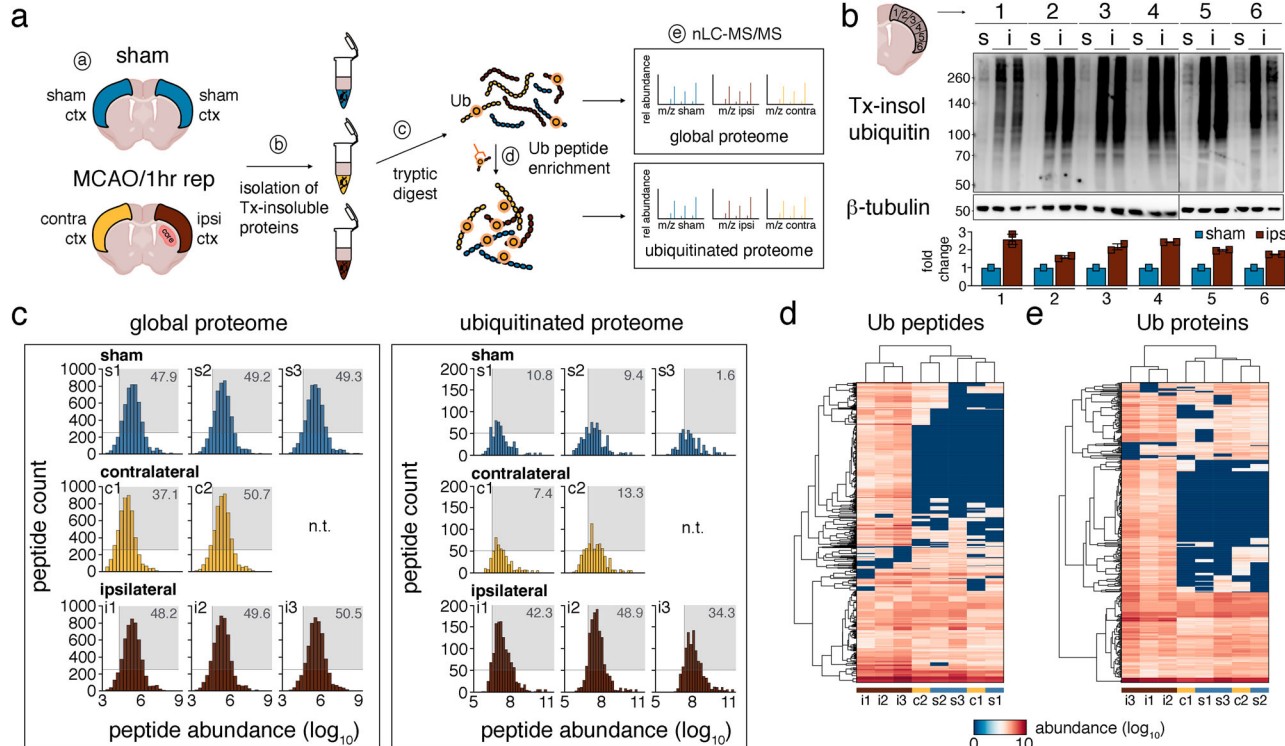

**Fig. 1 | nanoLC–MS/MS reveals increased protein ubiquitination in the ipsilateral cortex after cerebral ischemia. a** Schematic of the nanoLC–MS/MS experimental workflow. Mice underwent either sham or MCAO/1 h reperfusion surgeries, and sham, ipsi- and contralateral cortices were extracted (**a**). Triton-insoluble proteins were isolated (**b**) and digested with trypsin (**c**). Peptides with- and without K-ε-GG peptide enrichment (**d**) were subjected to nanoLC–MS/MS and downstream data analysis (**e**). *n* = 3 MS runs for sham and ipsilateral, *n* = 2 MS runs for contralateral (including each 20 mice/group). **b** Confirmation of increased protein ubiquitination after ischemia in detergent-insoluble fractions throughout the entire cerebral cortex. Ubiquitination was assessed by Western Blotting with anti-ubiquitin antibody in sham and ipsilateral cortical areas used for proteomic analyses. Antibody against β-tubulin

served as loading control. *n* = 1–2 mice/group. **c** Graphs showing the count and abundance of global and ubiquitinated peptides in sham (s1–3), ipsi- (i1–3), and contralateral (c1–2) Triton-resistant samples across MS experiments. Gray boxes indicate the percentage of peptide hits with counts greater than 250 per abundance bin higher than $log_{10} = 5$ (for global peptides) or greater than 50 per abundance bin higher than $log_{10} = 6$ (for Ub peptides). **d** Heatmap of hierarchical clustering of ubiquitinated peptide abundances across experimental groups and MS runs. **e** Heatmap of hierarchical clustering of ubiquitinated protein abundances across experimental groups and MS runs. c contralateral, ctx cortex, hr hour, i ipsilateral, K-ε-GG diglycyl-lysine, MCAO middle cerebral artery occlusion, n.t. not tested, rep reperfusion, s sham, Tx Triton X100, Ub ubiquitin.

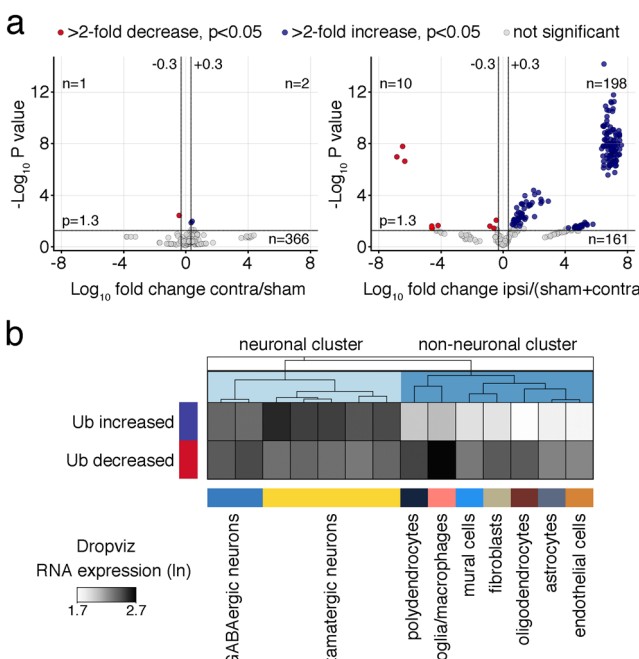

**Fig. 2 | Ischemic stroke leads to increased ubiquitination of 198 proteins with the highest expression levels in glutamatergic neurons. a** Volcano plots showing proteins with significantly changed ubiquitination in contralateral vs. sham groups (left panel) and ipsilateral vs. sham and contralateral control groups (right panel). Red circles indicate proteins with downregulated, blue circles with upregulated, and gray circles with not significantly changed ubiquitination. Data were pooled from 2 to 3 MS experiments. Significance at fold change = 2 with $P < 0.05$ (two-tailed unpaired $t$-test). **b** Heatmap showing the mean RNA expression levels (ln) of all proteins with increased or decreased post-ischemic ubiquitination in different brain cells in the frontal cortex. Single-cell mRNA expression levels were derived from the publicly available database Dropviz[27]. Ub ubiquitination.

proteins exhibited different abundance levels in contralateral vs sham samples (Fig. 2a), while 208 proteins were distinctly altered in ipsilateral to the stroke compared to control (sham + contralateral) samples (Fig. 2a). Among these, 198 (94.2%) showed increased and 10 (4.8%) decreased ubiquitination. Ubiquitin attaches to target proteins as chains with different architectures that regulate different cellular processes[26]. To gain initial insight into the possible functional consequences of increased ubiquitination after stroke, we determined ubiquitin lysine chain composition in control versus ipsilateral stroke samples. While lysine (K) 27-, K29-, and K33-chains were only minimally affected, K6-, K11-, K48-, and K63-chains were highly enriched after stroke (Supplementary Fig. 1), signifying changes in multiple ubiquitin-associated pathways. To evaluate whether changes in protein ubiquitination could be attributed to a specific cell type in the brain, we compared protein abundance levels with cortical single-cell sequencing RNA expression levels deposited in the Dropviz database[27] for qualitative cell-type assignment. This analysis indicated that proteins with downregulated ubiquitination were prominent in glial cells but not in neurons, whereas those with elevated ubiquitination were highly abundant in glutamatergic neurons and nearly absent in non-neuronal cells (Fig. 2b). In summary, we find increased ubiquitination of 198 predominantly neuronal proteins in the ipsilateral cortex after stroke.

### Post-ischemic protein ubiquitination is primarily found at the PSD

To further evaluate the subcellular localization of post-ischemic ubiquitinated proteins, we performed gene ontology (GO) enrichment analysis. In accordance with single-cell data, we found a significant association of

ubiquitinated proteins with neuronal compartments, such as the PSD, synapse, and postsynaptic membrane (Fig. 3a, Supplementary Fig. 2a). This connection was not found for proteins with decreased ubiquitination (Fig. 3a). In line with a synaptic localization of ubiquitinated proteins, both sucrose gradient- and detergent-isolated synaptic membrane- and PSD-preparations heavily stained for high molecular weight ubiquitin after ischemia (Fig. 3b, Supplementary Fig. 3). STRING analysis and k-means clustering visualized known protein-protein associations among ubiquitinated proteins and found two tight interaction clusters (Fig. 3c, depicted in yellow and blue) surrounded by two looser clusters (Fig. 3c, shown in green and red). The largest cluster 1 exclusively contained classic PSD-located proteins, including N-methyl-D-aspartate (NMDA)-receptor subunits Grin1 (GluN1), Grin2a (GluN2A), and Grin2b (GluN2B), alpha-amino-3-hydroxy-5-methyl-4-isoxazole-propionic acid receptor (AMPA)-receptor subunits Gria2 (GluA2) and Gria3 (GluA3), receptor tyrosine kinase (RTK) Ntrk2 (TrkB), scaffolding proteins Dlg2 (PSD93), Dlg4 (PSD95), and Shank2, as well as adapter proteins Dlgap1 (GKAP) and Dlgap2. A list of all ubiquitinated proteins grouped into functional families is provided in Supplementary Data 2. Elevated ubiquitination of PSD93, PSD95, Shank2, Shank3, GluN1, GluN2B, GluA2, and TrkB in ipsilateral MCAO-treated detergent-resistant lysates was further validated by immunoprecipitation (Fig. 3d). In summary, post-ischemic ubiquitination seems most prominent at the PSD.

### Ischemic stroke induces ubiquitination of PSD kinases

To assess what biological processes could be affected by post-ischemic ubiquitination, we again carried out GO enrichment analysis, which suggested a role in signal transduction and phosphorylation (Fig. 4a, Supplementary Fig. 2b). Proteins associated with these functions were again predominant at the PSD, where they were predicted to exhibit kinase activity (Fig. 4b, Supplementary Fig. 2c, d). STRING clusters confirmed a strong connection of post-ischemic ubiquitinated kinases with the PSD, as many are found in PSD-associated cluster 1 (e.g., Camk2a (CaMKIIα), Camk2b (CaMKIIβ), Camk2d (CaMKIIδ), Prkca (PKCα), Cdk5) and closely related cluster 2 (e.g., Prkcb (PKCβ), Prkcg (PKCγ), Ptk2b (Pyk2), Ckb (CKβ), Camk4 (CaMKIV), Itpkb (IP3KB), and AK1) (Fig. 3c). We biochemically confirmed post-ischemic ubiquitination of select PSD-localized kinases CaMKIIα, CaMKIIβ, PKCβ, PKCγ, Cdk5, Pyk2, CKβ, and the phosphatase Pten in detergent-resistant cortices by pull-down and detection of bound ubiquitin (Fig. 4c). We then performed staining with K48- and K63-ubiquitin linkage-specific antibodies to explore potential degradative and non-degradative functions of ubiquitin attachment to kinases after stroke. While CaMKIIα was modified with both ubiquitin chain types, neither was found attached to PKCβ (Supplementary Fig. 4). Although both Cdk5 and Pyk2 showed a tendency for modification with K63-ubiquitin, they were preferentially targeted by K48-ubiquitin (Supplementary Fig. 4). The number of ubiquitination sites identified by MS also greatly varied between kinases, with as little as three for Pyk2 and as many as 24 for CaMKIIβ (Fig. 4d, Supplementary Fig. 5). Importantly, many ubiquitination sites were found in kinase and regulatory domains (Fig. 4d), which led us to speculate that PSD-associated kinase activities could be affected by ubiquitin conjugation, as previously described for kinases involved in inflammatory and immune regulation[28,29].

### Ischemic stroke changes the PSD-associated phospho-proteome

To evaluate whether kinase activities could be altered at the PSD after ischemia on a global scale in an unbiased manner, we analyzed the phospho-proteome by nano-LC-MS/MS. As shown in Fig. 5a, b, concurrent with ubiquitination, we found a significant change in the phosphorylation of proteins after MCAO (listed in Supplementary Data 1, tab 4). As with ubiquitination, GO enrichment analysis indicated predominantly PSD-located phosphorylation changes (Fig. 5c). These were confirmed by a significantly increased detection of phospho-tyrosine in post-ischemic PSD vs

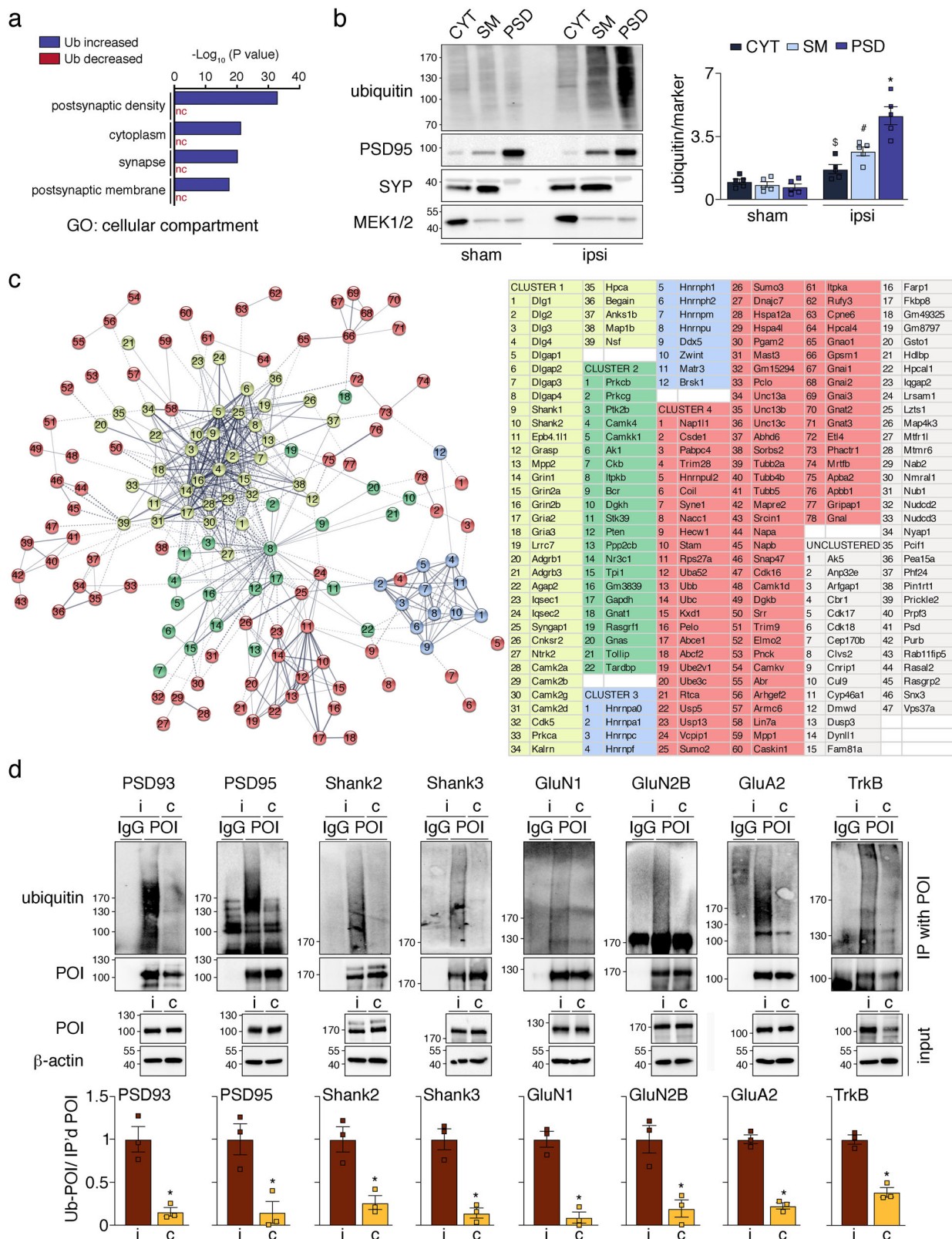

cytosolic or membrane fractions (Fig. 5d). Most proteins detected by MS exhibited increased phosphorylation, which, by antibody staining, was mainly attributed to enhanced tyrosine modifications (Fig. 5e). Global phosphorylation of serine/threonine residues, on the other hand, appeared unchanged (Fig. 5f). In conclusion, the data suggest that ischemia affects protein phosphorylation at the PSD.

**Ubiquitination reversibly suppresses activities of PSD-located CaMKII, PKC, and Cdk5**

Ischemia alters the activity of many prominent PSD kinases, such as CaMKII and PKC[30,31]. The underlying mechanisms, however, remain largely unknown. To determine if ubiquitination could be responsible for modified kinase activities and downstream protein phosphorylation after ischemia,

**Fig. 3 | Post-ischemic ubiquitination is primarily found at the PSD. a** GO enrichment analysis for cellular compartment of proteins with increased and decreased ubiquitination after ischemia. Benjamini-corrected *P* values for the four highest-ranking terms for the "Ub increased" dataset were plotted. **b** Detection of ubiquitin in cortical CYT, SM, and PSD fractions derived from mice that underwent sham or MCAO (ipsi) surgeries. Results were quantified. Western Blot with markers for CYT (MEK1/2), SM (SYP), and PSD (PSD95) was performed to show the purity of fractions. *n* = 5 mice/group; $^{\$}P$ = 0.0437 vs CYT sham; $^{\#}P$ = 0.0002 vs SM sham; *P* < 0.0001 vs PSD sham; two-tailed unpaired *t*-test. Data are expressed as mean ± s.e.m. **c** Proteins with increased ubiquitination after ischemia were analyzed for functional association networks using the STRING database. Out of 198, 151 proteins clustered into 4 k-means clusters (Cluster 1 yellow, Cluster 2 green, Cluster 3 blue, Cluster 4 red). A list of proteins found in each STRING cluster is provided in the same color code. Ubiquitinated proteins listed in gray were not directly associated with any cluster. **d** Biochemical verification of ubiquitination of select PSD-associated proteins by immunoprecipitation of the protein of interest from ipsi- and contralateral detergent-insoluble cortical fractions, and subsequent detection of ubiquitin by Western Blot. Precipitation with IgG-isotype antibodies served as controls. Results from *n* = 3 mice/group were quantified. PSD93: *P* = 0.0057; PSD95: *P* = 0.0183; Shank2: *P* = 0.0115; Shank3: *P* = 0.0031; GluN1: *P* = 0.0012; GluN2B: *P* = 0.0133; GluA2: *P* = 0.0003; TrkB: *P* = 0.0013. Two-tailed unpaired *t*-test. Data are expressed as mean ± s.e.m. c contralateral, CYT cytosol, GO gene ontology, i ipsilateral, IgG immunoglobulin G, IP immunoprecipitation, nc not called, POI protein of interest, PSD postsynaptic density, SM synaptic membrane, Ub ubiquitination.

we first assessed whether target phosphorylation of ubiquitinated CaMKII and PKC was affected by stroke. CaMKII and PKC both mediate phosphorylation of GluN2B S1303 and nNOS S847, which we found decreased after ischemia (Fig. 6a). In line with this, PSD-associated CaMKII and PKC activities were reduced as shown by less incorporation of radiolabeled phosphate from [γ-$^{32}$P]-ATP into respective kinase substrate peptides (Figs. 6b, c). To tie kinase activity reduction to ubiquitination, we exposed ipsilateral PSD lysates to the deubiquitinase USP2. Deubiquitination was confirmed by Western Blotting with an anti-ubiquitin antibody (Supplementary Fig. 6a). Removal of ubiquitin increased both CaMKII and PKC activities to pre-ischemic levels (Fig. 6d, e), corroborating that ubiquitination of these kinases reduces their kinase activities towards the tested substrates.

Another kinase with increased post-ischemic ubiquitination and affected by ischemia is Cdk5[32], a key kinase regulating synaptic plasticity and neuronal pathologies[33]. Cdk5 mediates serine phosphorylation of Tau S202/205 and Crmp2 S522 at the PSD, which was reduced after ischemia (Fig. 7a), accompanied by a decrease in Cdk5 activity (Fig. 7b). USP2 treatment of immunopurified Cdk5 from ipsilateral PSD lysates led to removal of ubiquitin (Supplementary Fig. 6b) and partial restoration of its activity (Fig. 7c), suggesting that ubiquitination is involved in Cdk5 activity suppression at the post-ischemic PSD. CKβ was another kinase showing reduced activity accompanied by elevated ubiquitination at the PSD after stroke (Fig. 4b, Supplementary Fig. 7a). However, de-ubiquitination was not effective in rescuing CKβ activity (Supplementary Figs. 7b, c), implying other activity-suppressing mechanisms in this case. To summarize, CaMKII, PKC, and Cdk5 exhibit reduced kinase activity at the PSD after ischemia, which can be reversed by the removal of ubiquitin.

## Ubiquitination reversibly activates PSD-associated Pyk2

Another kinase showing increased post-ischemic ubiquitination is Pyk2, a tyrosine kinase that was previously reported to exhibit aberrant function after ischemia[34]. In contrast to other tested kinases, we observed increased phosphorylation of Pyk2 target sites on GluN2B (Y1472), Src (Y419), and Tau (Y18) in ipsilateral cortical PSD fractions after MCAO, suggesting upregulation of PSD-associated Pyk2 activity (Fig. 8a). Indicative of increased Pyk2 activity, we detected elevated Pyk2 auto-phosphorylation on Y402 at 30 min after onset of reperfusion (Fig. 8b) as well as an increase in total Pyk2 tyrosine phosphorylation at 30 and 60 min after MCAO (Fig. 8c). To assess whether elevated Pyk2 activity was the result of ubiquitination, we used his-GST-Src as exogenous substrate. In line with increased Pyk2 activity after ischemia, his-GST-Src Y419 phosphorylation was elevated in samples exposed to Pyk2 isolated from mice that underwent MCAO when compared to sham (Fig. 8d). This increase was reversed in samples treated with USP2 (Fig. 8d, Supplementary Fig. 6c). In conclusion, ubiquitination is a reversible Pyk2 regulator after stroke.

## Discussion

In this study, we identified the ubiquitinated proteome in the ischemic periphery following ischemic stroke. We discovered that ubiquitinated proteins are particularly abundant at the PSD of glutamatergic neurons (Fig. 9a), constituting many PSD receptors, scaffolding proteins and their adapters, G proteins, and crucial effector and signaling proteins (Fig. 9b). Finally, we show that ubiquitination reversibly alters activities of the PSD-located kinases CaMKII, PKC, Cdk5, and Pyk2 (Fig. 9c), whose aberrant function is well-established to contribute to ischemic injury[30–32,34]. To our knowledge, these are the first data that implicate ubiquitination in the post-ischemic activity changes of critical kinases at the PSD.

The PSD is an electron-dense area on the postsynaptic side of excitatory synapses, which receives and propagates neurotransmitter signals from presynaptic terminals[35,36]. It contains a high number of scaffolding and signaling proteins that are physically and functionally connected to postsynaptic NMDA and AMPA receptors, thereby forming an intricate communication network whose tight control is paramount for neuronal wellbeing[36]. Biochemically isolated PSD fractions are rich in ubiquitin-immuno-positive conjugates[37–40], and ubiquitination of PSD-associated proteins critically regulates PSD function[39,41–43]. Albeit limited, there is previous evidence of elevated ubiquitination at the PSD after ischemia. In global brain ischemia caused by cardiac arrest, ubiquitinated proteins are found to be highly enriched at the PSD[44]. Additionally, proteomic analysis of ubiquitinated proteins after global ischemia detected PSD-enriched proteins such as CaMKII, GluN2A, GluN2B, GKAP, and PSD93, among others[45]. Notably, post-ischemic ubiquitinated proteins are resistant to detergent extraction and were therefore proposed to be aggregates[14,15]. Although this cannot be excluded for some proteins, our data indicate that the detergent inaccessibility is primarily due to the association of proteins with the inherently insoluble PSD[35].

Our study showed increased post-ischemic ubiquitination of many prominent PSD scaffolds, such as PSD93, PSD95, Shank2, Shank3, GKAP, and Dlgap2, as well as NMDA and AMPA receptor subunits. Ubiquitination of many of these proteins was reported to affect PSD function under physiological conditions. For example, ubiquitin conjugation to NMDAR and AMPAR controls receptor internalization, trafficking, recycling, and degradation, influencing synaptic plasticity and long-term potentiation[46–50]. GKAP and Shank ubiquitination induce structural changes at the PSD by modulating dendritic spine density and morphology[51,52]. PSD95 ubiquitination regulates AMPA receptor internalization, synapse elimination, PSD assembly, and plasticity via proteasome-dependent and independent mechanisms[42,53–55]. How the sharp increase in ubiquitination of these proteins after ischemia affects PSD assembly and function, however, remains to be determined. It could, for instance, underlie the increased thickening of the PSD found after ischemia, which is due to rearrangement of proteins, or the altered levels of PSD95 and CaMKII found at the PSD after ischemia and glutamate exposure[56,57]. Alternatively, it may induce the redistribution of NMDAR subunits found after ischemia or modify protein-protein interactions, such as seen for NMDAR subunits and PSD95[58,59].

Kinases were the largest enzymatic protein group to be ubiquitinated following stroke. The PSD is home to many kinases and phosphatases whose changing activity critically modulates PSD function, reshapes the structure of the PSD, and transmits signals to the neuronal cell body to evoke long-lasting and widespread effects on neuronal function[60–62]. Ubiquitination is well-established to affect kinase activity through degradative and non-degradative mechanisms, the latter by targeting regulatory regions

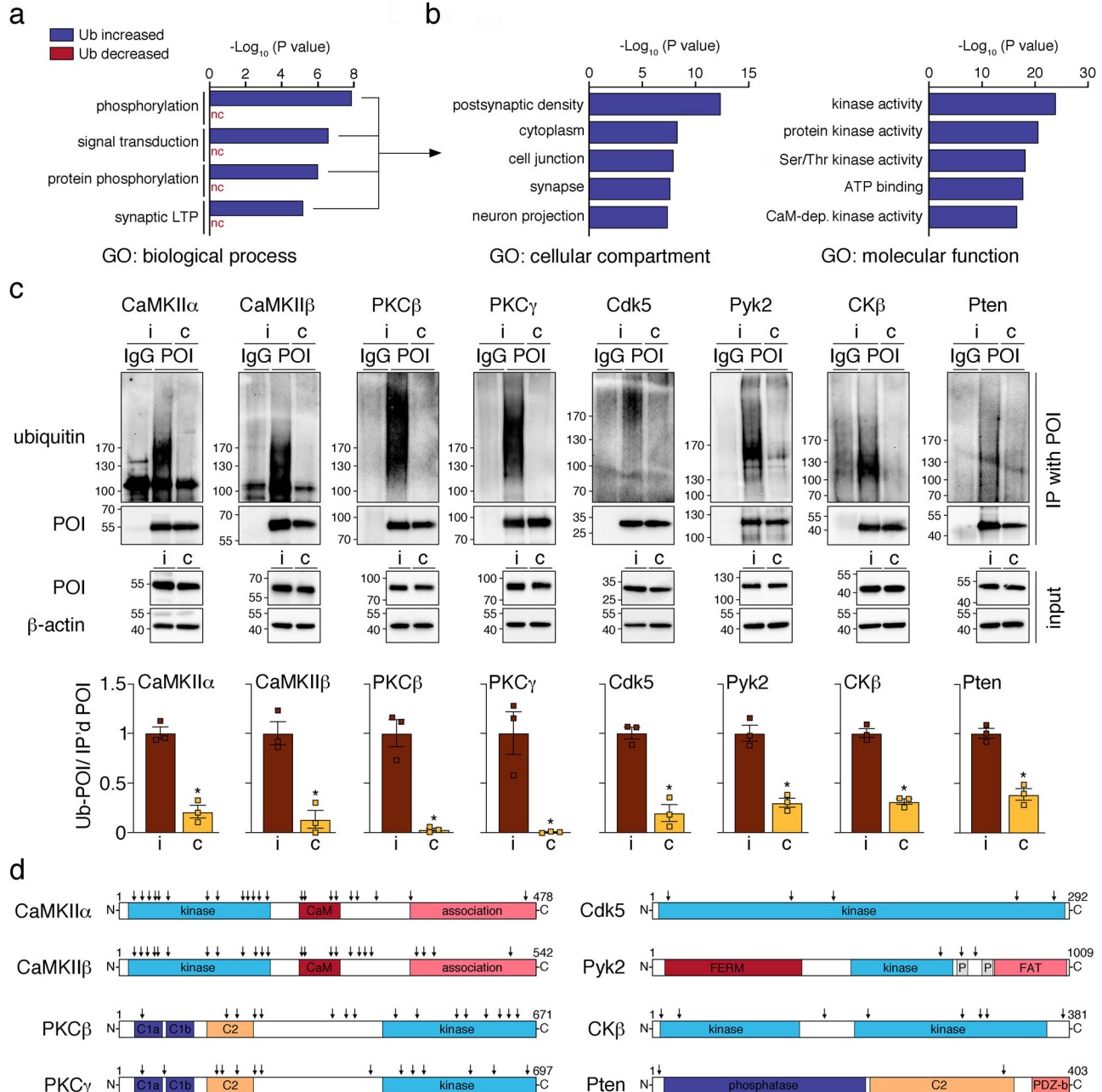

**Fig. 4 | Ischemic stroke results in ubiquitination of postsynaptic kinases and phosphatases. a** Proteins with increased and decreased post-ischemic ubiquitination were assessed for GO enrichment for biological process. Benjamini-corrected *P* values for the top four terms for the "Ub increased" dataset are depicted. **b** All proteins with increased ubiquitination after ischemia and enrichment in the four top categories for biological process were investigated for GO enrichment for molecular function and cellular components. Benjamini-corrected *P* values for the five highest-ranking terms are shown. **c** Post-ischemic ubiquitination of select PSD-associated kinases and phosphatases was confirmed by immunoprecipitation of proteins of interest from ipsi- and contralateral detergent-insoluble cortical lysates from MCAO/1 h reperfusion-treated mice and detection of ubiquitin by Western Blotting. IgG-isotype antibodies served as controls. Results from *n* = 3 mice/group were

quantified. CaMKIIα: *P = 0.0009; CaMKIIβ: *P = 0.0040; PKCβ: *P = 0.0021; PKCγ: *P = 0.0099; Cdk5: *P = 0.0015; Pyk2: *P = 0.0016; CKβ: *P = 0.0002; Pten: *P = 0.0013. Two-tailed unpaired *t*-test. Data are expressed as mean ± s.e.m. **d** Domain structure of prominent PSD-associated kinases and phosphatases found ubiquitinated after ischemia. Numbers represent amino acid positions, and arrows indicate ubiquitinated residues identified by MS analysis. BP biological process, c contralateral, C carboxy-terminus, C1a, and C1b diacylglycerol-binding domain, C2 calcium-binding domain, CaM calmodulin, FAT focal adhesion kinase-targeting domain, FERM, 4.1 protein, Ezrin radixin, and moesin domain, GO gene ontology, i ipsilateral, LTP long-term potentiation, N amino-terminus, nc not called, P proline-rich region, PDZ-b PSD95, Dlg1 Zo-1-containing domain-binding domain, POI protein of interest, Ub ubiquitination.

important for controlling kinase activity[63,64]. Ischemia disrupts the activity of many prominent PSD kinases, such as CaMKII, PKC, Cdk5, and Pyk2, exacerbating post-ischemic neuronal death[30–32,34]. Both CaMKII and PKC are inactivated by ischemia[31,65–68], whereas Cdk5 and Pyk2 show hyperactivation[32,34,69,70], but the underlying mechanism(s) have remained elusive. Here, we identified ubiquitin as a modifier of post-ischemic

CaMKII, PKC, Cdk5, and Pyk2 kinase activities. Staining with K48- and K63-ubiquitin linkage-specific antibodies revealed differential ubiquitination of these kinases, possibly implicating degradative and non-degradative mechanisms of regulation. Post-ischemic ubiquitination sites detected by nanoLC–MS/MS were concentrated in kinase, calcium-binding, or calmodulin-binding domains essential for the enzymatic activity of

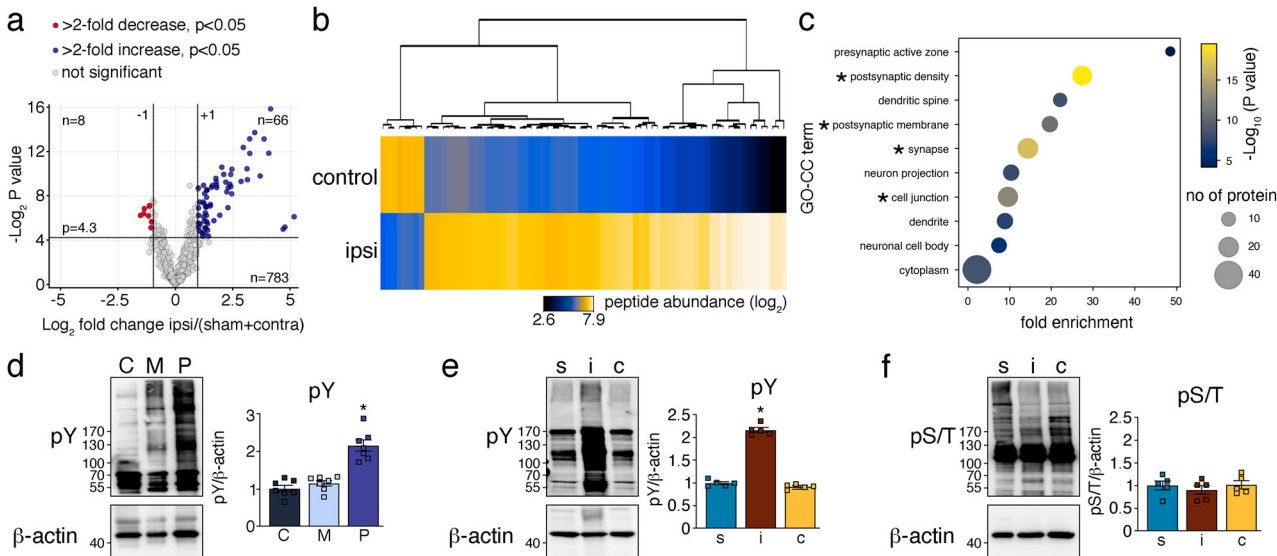

**Fig. 5 | Ischemic stroke alters protein phosphorylation at the PSD. a** The same homogenates used for ubiquitination studies were subjected to phospho-proteome analysis. The Volcano plot shows a significant increase and decrease of phosphorylation of 66 and 8 phospho-peptides, respectively, when comparing ipsilateral to control samples. Data were pooled from 2 to 3 MS experiments. Significance at fold change = 2 with $P < 0.05$ (two-tailed unpaired $t$-test). **b** Heatmap depicting hierarchical clustering of identified phospho-peptide abundances in control and ipsilateral samples. **c** GO analysis showing enrichment of proteins with changed post-ischemic phosphorylation at the PSD. Asterisks indicate terms with the most significant enrichment. **d** Cytosolic, synaptic membrane, and PSD-enriched fractions were obtained from ipsilateral cortices after MCAO/1 h reperfusion and stained for phospho-tyrosine. $n = 7$ animals/group; *$P < 0.0001$ from C and M; one-way ANOVA with Bonferroni test. **e** Cortical detergent-insoluble lysates from sham and MCAO-treated (ipsilateral and contralateral) animals were probed for the presence of phospho-tyrosine. $n = 5$ animals/group; *$P < 0.0001$ from s and c; one-way ANOVA with Bonferroni test. **f** Same as in **e**, but probed for phospho-serine/threonine. $n = 5$ animals/group. c contralateral, C cytosol, GO gene ontology, i ipsilateral, M membrane, P postsynaptic density, S serine, s sham, T threonine, Y tyrosine. Data are expressed as mean ± s.e.m.

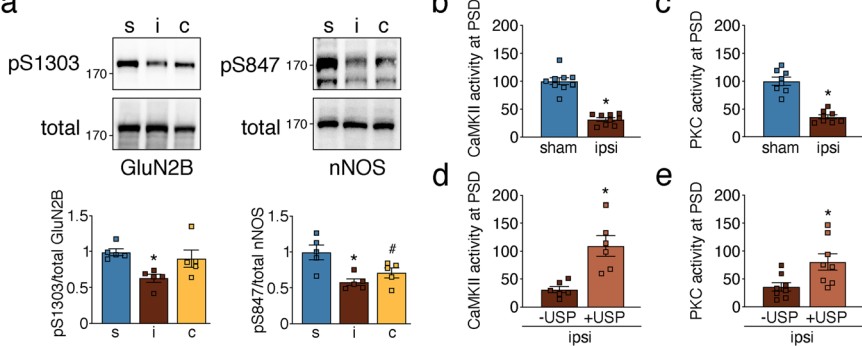

**Fig. 6 | Post-ischemic ubiquitination reversibly suppresses CaMKII and PKC activities at the PSD. a** Cortical PSD-lysates from sham and MCAO-treated mice were assessed for CaMKII and PKC downstream target phosphorylation. Phospho-specific vs total protein bands were quantified. GluN2B: *$P = 0.0162$ from s; nNOS: *$P = 0.0046$ from s, #$P = 0.0426$ from s; one-way ANOVA with Bonferroni test; $n = 5$ animals/group. **b** Cortical PSD-lysates from sham and MCAO-treated mice were assessed for CaMKII activity (*$P < 0.0001$, two-tailed unpaired $t$-test; $n = 9$ animals/

group). **c** Same as in **b**, but for PKC (*$P < 0.0001$, two-tailed unpaired $t$-test; $n = 8$ animals/group). **d** Post-ischemic PSD lysates were treated with recombinant deubiquitinase USP2, and CaMKII activity was reassessed and quantified (*$P = 0.0026$, two-tailed paired $t$-test; $n = 6$ animals). **e** Same as in **d**, but for PKC (*$P = 0.0012$, two-tailed paired $t$-test; $n = 8$ animals). c contralateral, i ipsilateral, S serine, s sham. Data are expressed as mean ± s.e.m.

respective kinases, and the removal of ubiquitin restored their activities. These data raise the possibility that ubiquitin could affect kinase function reversibly, making them potential therapeutic targets.

One discrepancy between previous studies and our work is the directionality of Cdk5 activity changes with stroke. In contrast to the published post-ischemic hyperactivation of Cdk5[32,70], we found reduced activity at the PSD. Since the majority of Cdk5 is located outside the PSD, this difference is of interest as it may indicate differential regulation in distinct subcellular compartments. In addition to kinases, we also observed ubiquitination of some phosphatases, such as Pten and PP2, supporting the conclusion that both phosphorylation and dephosphorylation at the PSD are affected in ischemia. Follow-up studies will

address the relationship between altered kinase and phosphatase activation, synaptic dysfunction, and neuronal death after ischemic stroke. In addition, since the activity of some ubiquitin ligases is regulated by phosphorylation[64], it would be interesting to address whether the observed changes in phosphorylation could affect post-ischemic protein ubiquitination. For example, Nedd4 and Vcpip1 exhibit increased phosphorylation after ischemia (Supplementary Data 1, tab 4), but whether this also affects their function is yet unclear.

In addition to classical PSD-associated proteins, we also identified several RNA-binding proteins, chaperones, and proteins of the ubiquitination cascade to be ubiquitinated post-ischemia. Many of the ubiquitinated proteins belonging to these classes, including ubiquitin ligases Trim9 and

Ube3c, deubiquitinases Usp5, Usp13, and Vcpip1, the ubiquitin conjugase Uev-1, Sumo2/3, as well as heat shock proteins Hspa12a and Hspa4l, are also present at the PSD[71–76]. Whether these are located at the PSD at the time of ubiquitination, however, needs to be determined. Sumo2/3 was previously identified as a ubiquitination target after ischemia[25], which we verified here.

The RNA-binding proteins TDP43 and hnRNPA1 translocate from a soluble to detergent-insoluble state after ischemia[77], which may represent an association with RNA granules or aggregates[78]. On the other hand, it may indicate that these RNA-binding proteins travel to the PSD to regulate localized protein synthesis[79,80]. These possibilities need further exploration.

Other protein classes with high ubiquitination levels after ischemia included G-proteins, GTP regulators, cytoskeletal proteins, proteins involved in cellular trafficking and transport, as well as metabolic enzymes. How post-ischemic ubiquitination alters their function is unclear. One of the GTPase regulators localized to the PSD, SynGAP1, is ubiquitinated and degraded after ischemia, which aggravates ischemic injury[81]. Notably, our study confirmed increased post-ischemic ubiquitination of SynGAP1.

In summary, our study identified PSD-localized proteins as major ubiquitination targets after cerebral ischemia (Fig. 9a, b). Particularly, we discovered that ubiquitination is a potent modulator of ischemia-relevant neuronal kinases at the PSD (Fig. 9c). This finding shifts the role of post-ischemic ubiquitination away from protein aggregation to reversible regulatory modification of critical kinases involved in ischemic injury that may be targetable for stroke therapy.

## Methods
### Transient middle cerebral artery occlusion (tMCAO) model of focal ischemia
All animal procedures were approved by the Weill Cornell Medicine Institutional Animal Care and Use Committee (IACUC) and were carried out according to IACUC, NIH, and ARRIVE guidelines (http://www.nc3rs.org/ARRIVE). We have complied with all relevant ethical regulations for animal use. Transient focal cerebral ischemia was induced in 8–12 weeks C57BL6/J WT male mice (Jackson Laboratories, Bar Harbor, ME) via MCAO using an intraluminal filament, as described[14,82]. Briefly, mice were anesthetized with isoflurane, and the middle cerebral artery was occluded by

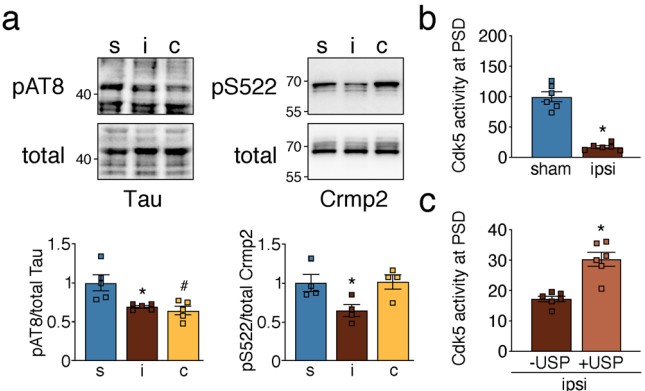

**Fig. 7 | Post-ischemic ubiquitination reduces PSD-associated Cdk5 activity.**
**a** Phosphorylation levels of Cdk5 downstream targets were determined in PSD lysates of sham and MCAO-treated animals. Tau: *$P = 0.0137$ from s, *$P = 0.0055$ from s; Crmp2: *$P = 0.0496$ from s and $P = 0.0430$ from c; one-way ANOVA with Bonferroni test; $n = 4–5$ animals/group. **b** Cortical PSD-lysates from sham and MCAO-treated mice were assessed for Cdk5 activity (*$P < 0.0001$, two-tailed unpaired $t$-test; $n = 6$ mice/group). **c** Cdk5 activity was measured in post-ischemic PSD lysates untreated and treated with recombinant deubiquitinase USP2 (*$P = 0.0057$, two-tailed paired $t$-test; $n = 6$ animals). c contralateral, i ipsilateral, S serine, s sham. Data are expressed as mean ± s.e.m.

**Fig. 8 | Ubiquitination reversibly activates PSD-located Pyk2 after ischemia. a** Phosphorylation of Pyk2 kinase target proteins was analyzed in PSD extracts. GluN2B: *$P = 0.0016$ from s and *$P = 0.0001$ from c; Src: *$P < 0.0001$ from s and *$P = 0.0003$ from c; Tau: *$P < 0.0001$ from s and c; one-way ANOVA with Bonferroni test; $n = 5$ animals/group. **b** Pyk2 tyrosine phosphorylation was assessed on Y402 (*$P = 0.0184$ from s; **$P = 0.0040$ from s and **$P < 0.0001$ from 30 min rep; one-way ANOVA with Bonferroni test; $n = 4–6$ animals/group). **c** Overall Pyk2 tyrosine phosphorylation was assessed after Pyk2 pull down with a phospho-tyrosine-specific antibody (*$P = 0.0089$ from s; **$P = 0.0005$ from s; one-way ANOVA with Bonferroni test; $n = 4-6$ animals/group).
**d** Phosphorylation of immobilized recombinant his-GST-Src by Pyk2 derived from sham and MCAO lysates untreated or treated with recombinant USP2 was determined. *$P = 0.0152$ from s; #$P = 0.0242$ from i/untreated; one-way ANOVA with Bonferroni test; $n = 4$ animals/group. c contralateral, i ipsilateral, S serine, s sham, Y tyrosine. Data are expressed as mean ± s.e.m.

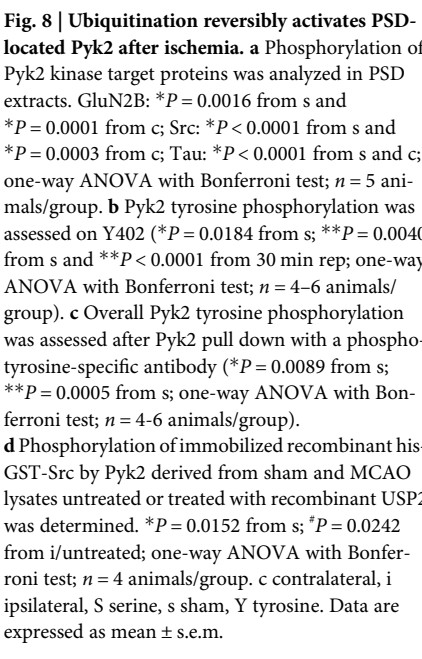

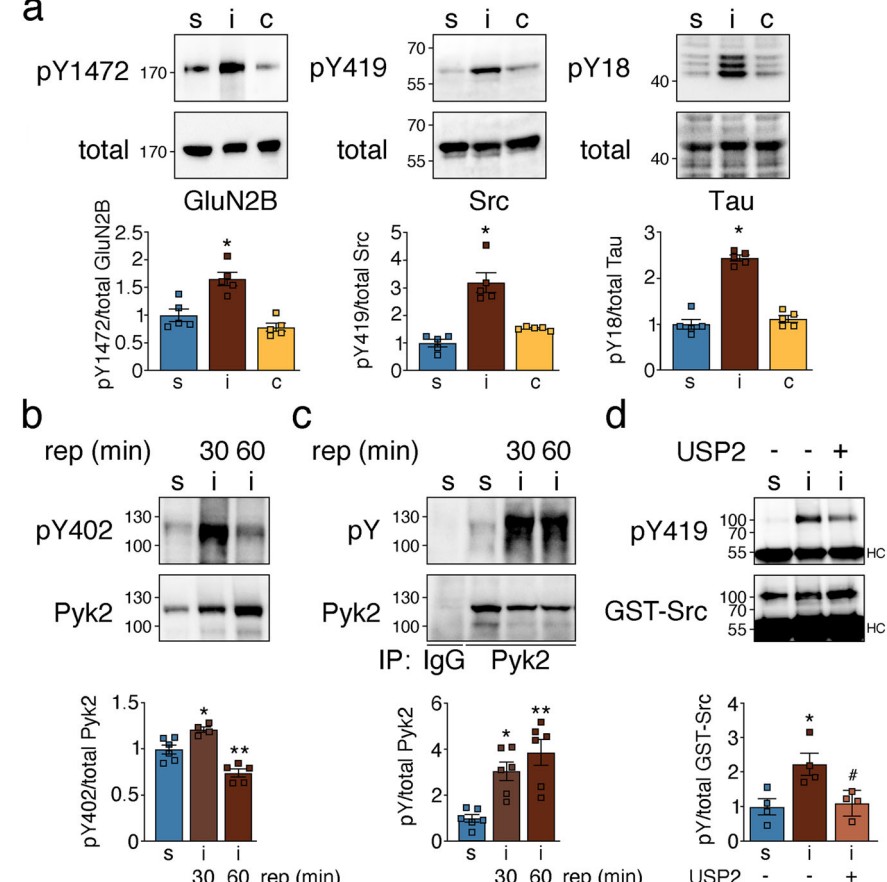

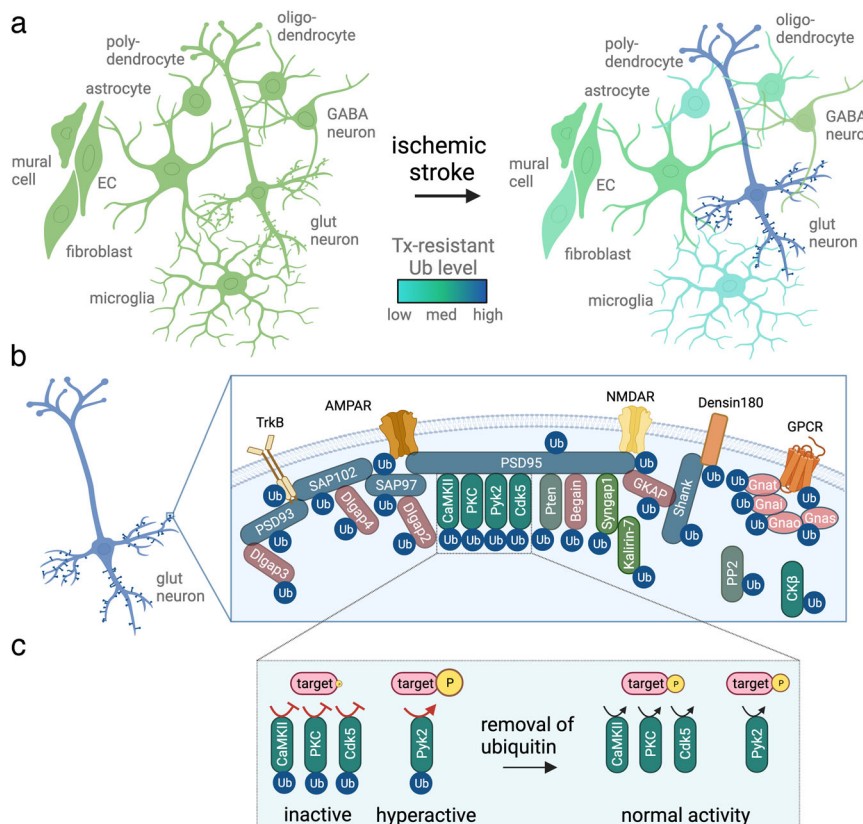

**Fig. 9 | Summary of the findings of this study. a** Post-ischemic detergent-resistant ubiquitination is elevated in neurons, particularly at the postsynaptic density (PSD) of glutamatergic neurons, while it is reduced in all other brain cell types. EC endothelial cell, glut glutamatergic, Tx Triton X100, Ub ubiquitination. **b** Postsynaptic proteins with increased ubiquitination after ischemic stroke include receptors (alpha-amino-3-hydroxy-5-methyl-4-isoxazole-propionic acid receptor (AMPAR), N-methyl-D-aspartate receptor (NMDAR), Densin-180, G protein-coupled receptor (GPCR), tropomyosin receptor kinase (TrkB)), scaffolding proteins and their adapters (brain-enriched guanylate kinase-associated protein (Begain), Disks large-associated protein 2–4 (Dlgap2-4), guanylate kinase-associated protein (GKAP), postsynaptic density protein 93/95 (PSD93/95), synapse-associated protein 97/102 (SAP97/102), SH3 and multiple ankyrin repeat domain (Shank)), G proteins (guanine nucleotide-binding protein G(i) subunit alpha (Gnai), guanine nucleotide-binding protein G(o) subunit alpha (Gnao),

guanine nucleotide-binding protein G(s) subunit alpha isoforms short (Gnas), guanine nucleotide-binding protein G(t) subunit alpha (Gnat)), and signaling proteins (calcium-calmodulin-dependent protein kinase II (CaMKII), cyclin-dependent kinase 5 (Cdk5), creatine kinase b (CKb), Kalirin-7, protein kinase C (PKC), protein phosphatase 2 (PP2), phosphatase and TENsin homolog (Pten), proline-rich tyrosine kinase (Pyk2), synaptic Ras GTPase-activating protein 1 (SynGAP1)). **c** PSD-kinases, such as CaMKII, PKC, Cdk5, and Pyk2, are prominent post-ischemic ubiquitination targets. While CaMKII, PKC, and Cdk5 activities at the PSD are decreased after stroke, leading to reduced target phosphorylation, Pyk2 exhibits increased activity, thereby accelerating the phosphorylation of target proteins. In all cases, kinase activity regulation after stroke was dependent on ubiquitination, whose removal normalized activity. P phosphorylation. This figure was created with BioRender.com.

insertion of either a homemade heat-blunted 6-0 nylon- or a purchased 6-0 fine silicone rubber-coated-monofilament (Doccol Corporation, Sharon, MA) into the right external carotid artery and further progression through the internal carotid artery. Simultaneously, the right common carotid artery was ligated. To induce ischemia, the filament remained in place for 30–40 min, after which reperfusion was initiated by its removal. Transcranial laser Doppler flowmetry (Periflux System 5010; Perimed Inc, Ardmore, PA) was used to monitor cerebral blood flow (CBF). Only mice with >85% reduction in CBF during the ischemic period and >80% increase after 10 min of reperfusion were included in this study. For sham surgery, vessels were visualized and cleared of connective tissue with no further manipulations. In all mice, body temperature was maintained at 37 °C during the procedure. After a 1 h reperfusion period, mice were sacrificed, and brains were removed. Four or six-mm cortical slices (4 mm: +2.25 to −1.76 mm bregma; 6 mm: +3.25 to −2.76 mm bregma) were dissected from MCAO and control regions using a brain matrix (Kent Scientific, Torrington, CT). Samples were either flash-frozen in liquid N$_2$ and stored at −80 °C until further processing (MS, ubiquitination, and phosphorylation studies), or used fresh (kinase activity measurements).

### Isolation of Triton-insoluble proteins from cortical tissue for MS and biochemical analyses
The Triton-insoluble fraction containing ubiquitinated proteins was isolated from cortices as described[14,77]. Cortical tissue was mechanically homogenized in glass douncers in 500 μL ice-cold buffer/6 mm tissue (15 mM Tris pH7.5, 1 mM MgCl$_2$, 250 mM sucrose, 2.5 mM EDTA, 1 mM EGTA, 1 mM dithiothreitol (DTT), 20mM N-ethylmaleimide, 1× protease- and 1× phosphatase-inhibitor cocktail (Roche Applied Science, Penzberg, Germany)). Homogenates were spun at 10,000 rpm for 10 min at 4 °C to obtain the supernatant representing the cytosolic fraction. The resulting pellets were sonicated in the 350 μL homogenization buffer at 4 °C for 10 s at 20% amplitude (Branson Ultrasonics, Danbury, CT). Finally, Triton X100 and KCl were added to an end-concentration of 2% and 150 mM, respectively, and lysates were incubated for 1 h rotating at 4 °C. Triton-soluble fractions, containing membranes and -insoluble pellets were obtained by centrifugation at 10,000 rpm for 10 min at 4 °C.

### Ubiquitinome analysis
**Protein digestion and enrichment of ubiquitinated peptides.** Triton-insoluble pellets obtained from 20 cortices were pooled and solubilized by

sonication in 50 mM phosphate buffer pH7, 7 M urea, 2 M thiourea, and 0.1% SDS. Protein concentration was determined by Bradford assay using BSA as the calibrant and further confirmed by running samples on a precast NOVEX 10% Bis–Tris mini-gel (Thermo Fisher Scientific, Waltham, MA) alongside a serial dilution of an *E. coli* lysate (2.5, 5, 10, 15 µg/lane). Protein bands were visualized with colloidal Coomassie blue stain (Thermo Fisher Scientific) and quantified with a Typhoon 9400 scanner by ImageQuant Software version TL 8.1 (GE Healthcare, Chicago, IL). 3.6 mg of protein per sample was reduced with 10 mM Tris(2-carboxyethyl) phosphine for 1 h at 34 °C, alkylated with 23 mM iodoacetamide for 1 h in the dark, and then quenched with a final concentration of 27 mM DTT. Samples were diluted with 5 volumes 25 mM Tris pH8, exposed to 72 µg trypsin (1:50 v/v) for 16 h at 37 °C, and cleaned up by loading on a preconditioned SCX PolySULFOETHYL Aspartamide cartridge (10 mm i.d. × 14 mm, 12 µm particle size, 300 Å pore size; PolyLC Inc., Columbia, MD) and washing with 2 mL buffer containing 10 mM potassium phosphate ($K_3PO_4$) pH3 and 25% acetonitrile (ACN). Peptides were eluted from the cartridge with 1 mL buffer solution (10 mM $K_3PO_4$ pH3, 25% ACN, and 500 mM KCl), desalted through solid phase extraction (SPE) on Sep-Pak cartridges (Waters, Milford, MA), and evaporated to dryness in a speed vacuum device. Enrichment for ubiquitinated peptides was conducted using a PTM Scan® Ubiquitin Remnant Motif [K-ε-GG] kit (Cell Signaling, Danvers, MA) following the vendor's recommended procedures. Specifically, peptides were reconstituted in 350 µL immunoaffinity purification (IAP) buffer, transferred to a vial containing 20 µL equilibrated Ubiquitin Remnant motif antibody beads, and incubated on a vortex mixer at 4 °C for 2 h. After centrifugation at 2000*g* for 30 s, the beads were washed twice with 250 µL IAP buffer and three times with ddH2O. Finally, the enriched peptides were eluted three times with each 55 µL 0.15% TFA. The eluted fractions were pooled, dried, and reconstituted in 22 µL 0.5% formic acid (FA) for subsequent label-free quantitative analysis by nanoLC–MS/MS.

### nanoLC–MS/MS

The nanoLC–MS/MS analysis was carried out using an Orbitrap Fusion mass spectrometer (Thermo Fisher Scientific) equipped with a nanospray Flex Ion Source with high energy collision dissociation (HCD) as described[83,84] and coupled with UltiMate3000 RSLCnano (Dionex, Sunnyvale, CA). Reconstituted ubiquitinated peptides (20 µL/sample) were injected onto a PepMap C-18 RP nano trap column (3 µm, 100 µm × 20 mm, Dionex) with nanoViper fittings at 20 µL/min flow rate for on-line desalting, separated on a PepMap C-18 RP nano column (3 µm, 75 µm × 25 cm), and eluted in a 120 min gradient of 5–35% ACN in 0.1% FA at 300nL/min, followed by a 8-min ramping to 95% ACN in 0.1% FA, and a 9-min hold at 95% ACN in 0.1% FA. The column was re-equilibrated with 2% ACN in 0.1% FA for 25 min prior to the next run. The Orbitrap Fusion was operated in positive ion mode with nano spray voltage set at 1.6 kV and source temperature at 275 °C. External calibration for Fourier Transform (FT), Ion Trap (IT), and quadrupole mass analyzers was performed. The instrument was operated in data-dependent acquisition (DDA) mode using FT mass analyzer for the initial survey MS scan for selecting precursor ions. This was followed by 3 s "Top Speed" data-dependent collision-induced dissociation (CID) ion trap MS/MS scans for precursor peptides with 2–7 charged ions above a threshold ion count of 10,000 with a normalized collision energy of 30%. MS survey scans had a resolving power of 120,000 (full width at half maximum [FWHM] at m/z 200) for the mass range of m/z 400–1600 with an automated gain control (AGC) of $3 \times 10^5$ and a max IT of 50 ms. Dynamic exclusion parameters were set to 1 within 45 s exclusion duration with ±10ppm exclusion mass width. Data were acquired under Xcalibur 3.0 operation software and Orbitrap Fusion Tune 2.0 (Thermo Fisher Scientific).

### Phospho-proteomics

**Protein digestion and TMT8-plex labeling.** In total, 200 µg protein per sample was dissolved in 50 mM triethylammonium bicarbonate (TEAB)

pH 8.5, 7 M urea, 2 M thiourea, and 1% SDS, and reduced, alkylated, and quenched as described above. After quenching, phosphoric acid (PA) was added to samples at a final concentration of 1.2%, followed by 1:7 dilution (v/v) with 90% methanol in 100 mM TEAB pH 8.5. Samples were loaded onto S-Trap Mini Spin columns (Protifi, Huntington, NY) for subsequent S-trap-based tryptic digest[83]. Columns were centrifuged at 4000 g for 30 s, and washed three times with 400 µl 90% methanol in 100 mM TEAB pH 8.5. Digestion was performed by adding 100 µL trypsin (80 ng/µL; 1:25 w/w) in 50 mM TEAB pH 8.5 onto the spin columns. The solution was absorbed into the highly hydrophilic matrix and incubated for 16 h at 37 °C. The digested peptides were eluted from the S-trap column sequentially with 100 µL of each 50 mM TEAB pH8.5, 0.2% FA, and 50% ACN in 0.2% FA. Eluates were pooled and dried in a speed vacuum device for subsequent TMT8-plex labeling. Labeling was performed according to Thermo Fisher Scientific's TMT Mass Tagging Kits and Reagents protocol with slight modifications as described in[84–86]. Briefly, dried samples were resuspended in 100 µL ddH2O, re-dried, and reconstituted in 100 µL 100 mM TEAB pH 8.5. The TMT8-plex labels (0.8 mg dried powder) were reconstituted in 50 µL anhydrous ACN and added at a 1:2 ratio to each sample for 1 h at room temperature. Labeled peptides from each dataset were pooled, evaporated to dryness, and subjected to cleanup by SPE on MCX cartridges (Waters). Incorporation of the TMT label was verified using Orbitrap Fusion (Thermo Fisher Scientific). Two aliquots, each containing 600 µg labeled peptides, were used for the enrichment of phospho-peptides with titanium dioxide (TiO2) and ferric nitrilotriacetate (Fe-NTA), respectively.

**Enrichment of phospho-peptides with TiO2 beads.** TiO2 enrichment was conducted using a TiO2 Mag Sepharose kit (GE Healthcare)[85,86]. Essentially, 600 µg aliquots of TMT8-plex-tagged tryptic peptides were reconstituted in 400 µL buffer containing 1 M glycolic acid, 80% ACN, and 5% trifluoroacetic acid (TFA) and incubated with 100 µL TiO2 beads for 30 min with 1800 rpm vortex. After washing the beads with 80% ACN and 1% TFA, the phospho-peptides were eluted 3 times with each 200 µL 5% ammonium hydroxide. The eluted fractions were pooled, filtered through a Spin-X column (22 µm pore size; Corning Inc., Corning, NY), dried, and reconstituted in 50 µL 0.5% FA for subsequent nanoLC–MS/MS analysis.

**Enrichment of phospho-peptides with Fe–NTA beads.** Fe-NTA enrichment was carried out with a High Select Fe-NTA phospho-peptide enrichment kit (Thermo Fisher Scientific), as described[85,86]. TMT8-plex-tagged tryptic peptides (600 µg) were reconstituted in binding/washing buffer (80% Acetonitrile, 1% TFA). The reconstituted peptides were then applied to a spin column containing equilibrated Fe–NTA beads and incubated at room temperature for 30 min. After washing the beads with binding/washing buffer, phospho-peptides were eluted 3 times with each 100 µL elution buffer (5% ammonium hydroxide). Pooled eluates were filtered through a Spin-X column (22 µm pore size; Corning Inc.), dried, and reconstituted in 50 µL 0.5% FA for nanoLC–MS/MS analysis.

### nanoLC–MS/MS

Fractions enriched by both TiO2 and Fe-NTA were pooled and analyzed with the same HPLC–MS instrumentation as described above. The Orbitrap Fusion instrument was operated in both multiple-stage activation (MSA)-based Synchronous Precursor Selection (SPS) MS3 method and neutral loss (NL) triggered MS3 method for phospho-peptides under 3 s "Top Speed" data-dependent analysis, as reported[86,87]. In the MSA SPS MS3 method, the Orbitrap analyzer was used for acquiring CID MS2 spectra to increase confidence in phospho-peptide identifications and site location. The multiple MS2 fragment ions isolated by SPS were then subjected to HCD fragmentation at the MS3 level for reporter ion quantitation. In the NL-triggered MS3 method, a regular MS2 spectrum was first generated from the precursor using CID. If an NL peak was detected, then an HCD MS3 scan was acquired on the

most abundant ion to obtain both the sequence information and quantitation.

## Global proteomics

Protein digestion and TMT8-plex labeling were performed as laid out in the phospho-proteomics section. An aliquot containing 400 μg labeled peptides was used for first-dimensional LC fractionation via high pH reverse phase (hpRP) chromatography.

### High pH reverse phase chromatography.

hpRP chromatography was performed as reported using a Dionex UltiMate 3000 HPLC system with a built-in micro-fraction collector and UV detection (Thermo Fisher Scientific)[88]. Briefly, 400 μg TMT8-plex-tagged tryptic peptides were reconstituted in buffer A (20 mM ammonium formate ($NH_4FA$) pH 9.5) and loaded onto a XBridge® C18 column (3.5 μm, 2.1 × 150 mm; Waters). LC was performed using a gradient of 10–45% buffer B (85% ACN and 15% 20 mM $NH_4FA$) for 30 min at a flow rate of 200 μL/min. Forty-eight fractions were collected at 1 min intervals and pooled into 10 fractions based on UV absorbance at 214 nm using a multiple fraction concatenation strategy. Each of the 10 fractions was dried and reconstituted in 100 μL 2% ACN in 0.5% FA for nanoLC–MS/MS analysis.

### nanoLC–MS/MS.

The nanoLC–MS/MS analysis was performed with the same Orbitrap Fusion system described above but using high energy collision dissociation (HCD) as published[85,86]. NanoLC parameters were the same as for ubiquitinated peptides with a 120 min gradient. The Orbitrap Fusion was operated in data-dependent acquisition (DDA) mode using an FT mass analyzer for one survey MS scan to select precursor ions, followed by 3 s "Top Speed" data-dependent HCD–MS/MS scans for precursor peptides with 2–7 charged ions, above a threshold ion count of 20,000 with normalized collision energy of 40%. MS survey scans had a resolving power of 120,000 (FWHM at m/z 200) for the mass range of m/z 375-1600 with an AGC of $4 \times 10^5$ and a max IT of 50 ms, and MS/MS scans at 60,000-resolution with an AGC of $1 \times 10^5$, max IT of 86 ms, while the Q isolation window was 1.6Th for the mass range m/z 105–2000. Dynamic exclusion parameters were set to 1 within 50 s exclusion duration with ±10ppm exclusion mass width. Data were acquired under Xcalibur 3.0 operation software (Thermo Fisher Scientific) in Orbitrap Fusion.

## MS data analysis and peptide quantitation

Global, phospho-, and ubiquitin-enriched MS-raw data were analyzed by MaxQuant version 1.6.0.1 (Max Planck Institute of Biochemistry, Martinsried, Germany)[89] and Proteome Discoverer-Sequest (PD) 2.2. algorithms against the UniProt mouse database (UP000000589_10090, download 2018), respectively. Default settings for TMT8-plex reporter-ion quantitation were used for global proteins and phospho-peptide abundances. Ubiquitinated peptide abundances were determined by the label-free quantification (LFQ) MaxQuant algorithm. Two-missed cleavages by trypsin were allowed. MS-peptide mass tolerance was set to 10ppm and MS/MS tolerance to 0.03 Da and 0.6 Da for global, phospho-, and ubiquitin enrichment raw data analysis, respectively. For analysis of TMT-labeled global proteome measurements, methionine oxidation and acetylation of amino-termini were specified as variable modifications, whereas cysteine carbamidomethylation, lysine TMT6-modifications and amino-terminal amines (for TMT labeled files) were specified as fixed modifications. For quantitative phospho-peptide analysis, phosphorylation on serine (S), threonine (T), and tyrosine (Y) residues were additionally specified as variable modifications. Estimated false discovery rate (FDR) thresholds for protein, peptide, and modification sites were specified at a maximum 1%. Minimum peptide length was set at 6, and all other parameters in MaxQuant were set to default values[89]. For analysis of ubiquitin-enrichment samples in MaxQuant, variable modification settings were changed to glycine–glycine at lysine, N-ethylmaleimide at cysteine, oxidation at methionine and acetylation of amino-termini. To confidently identify phosphorylation sites, the

phosphoRS 3.0 node, which determines individual probability values for each putatively phosphorylated site within the validated peptide sequences[90], was integrated into the PD 2.2 workflow.

For further analysis, all intensity values were transformed to a log scale, and only high-confidence proteins that appeared in all MS runs were considered. For the ubiquitome, we detected the following number of peptides/proteins: run 1, 1664/709; run 2, 1803/763; run 3, 1220/529. Out of these, 369 proteins (with 1784 ubiquitinated and 5282 global peptides) were found in all 3 runs and further analyzed. For the TMT phospho-proteome, across all runs, 1244 peptides belonging to 400 proteins were detected. Out of these, 857 peptides matching to 322 proteins were found in all 3 runs and included in the analysis. Peptide abundances from mass-spectrometry runs (40–60 mice total/group) were median averaged. A list of identified high-confidence peptides and proteins can be found in Supplementary Data 1. Raw data have been deposited to the ProteomeXchange Consortium (http://proteomecentral.proteomexchange.org) via the PRIDE partner repository with the dataset identifier PXD042951. Relative phosphorylated and ubiquitinated peptide abundance levels were calculated for sham vs. contralateral stroke and ipsilateral stroke vs. control (sham/contralateral stroke). Protein phosphorylation was assessed at the peptide level only, whereas ubiquitination was determined at peptide and protein levels. For the latter, peptide levels for each protein were averaged. Due to their overlapping ubiquitination patterns, we used an average control value derived from sham + contralateral samples to evaluate changes in the ipsilateral group. Outlier analysis and exclusion were executed by the interquartile range (IQR) method (±1.5× IQR). Significantly changed ubiquitinated and phosphorylated proteins were defined as ±2-fold change with a $P < 0.05$.

Ubiquitinated proteins identified by MS were analyzed for functional interpretation using the following tools. Abundance binning was performed with the Perseus software[91]. The single-cell sequencing database Dropviz[27] was employed to determine the brain cell type with the highest expression of ubiquitinated proteins. Dropviz datasets used were: Astrocyte_Gja1, Endothelial_Flt1, Fibroblast-Like_Dcn, Interneuron_CGE_Cplx3-Synpr, Interneuron_MGE_Sst-Pvalb, Microglia_Macrophage_C1qb, Mural_Rgs5Acta2, Neuron_Layer6Subplate_Syt6, Neuron_Layer23_Nptxr, Neuron_Layer5b_Fezf2, Neuron_Claustrum_Nr4a2, Neuron_Layer5_Parm1, Oligodendrocyte_Tfr, Polydendrocyte_Tnr. Protein interaction network analysis was performed using interaction data (text mining and experiments) from the Search Tool for the Retrieval of Interacting Genes/proteins (STRING) database[92]. Only interactions with a confidence setting of 0.4 are represented in the network. To find interaction groups in the dataset, clusters were identified with k-means clustering (4 clusters). GO enrichment analysis was performed using the functional annotation tool of the DAVID bioinformatics resources (Version 6.8)[93]. Enriched terms were sorted by the Benjamini-Hochberg adjusted P value. Results were visualized with Python-based software platforms Instant Clue (Version 0.10.10, University of Cologne, Germany)[94] and Orange3 (Version 3.35.0, University of Ljubljana, Slovenia), as well as GraphPad Prism (Version 9.5.1).

## Isolation of the postsynaptic density by sucrose gradient centrifugation

The postsynaptic density (PSD) from cortical tissue was isolated as described[35], with some modifications. All buffers were supplemented with 1 mM sodium orthovanadate, 5 mM glycerol 2-phosphate, and 1× ProBlock Gold Protease Inhibitor Cocktail (GoldBio, St. Louis, MO), and cooled to 4 °C before use. Two 6-mm cortical slices were homogenized in 5 mL 25 mM Tris pH7.5/0.3 M sucrose (TS buffer) in a dounce homogenizer. An additional 2 mL TS buffer was added, and the mixture was spun at 1500g for 10 min at 4 °C. The supernatant was carefully decanted into a new tube and placed on ice. The pellet was re-homogenized in TS buffer and spun again to remove nuclei. The supernatants were then combined and centrifuged at 17,000 g for 10 min at 4 °C in a Beckman SW41-Ti rotor (Beckman Colter, Brea, CA). The supernatant was saved as the cytosolic (CYT) fraction. The pellet was resuspended in 2 mL 25 mM Tris pH 7.5 by homogenization in a

dounce homogenizer and incubated on ice for 20 min. The homogenized fraction was layered onto a 0.8–1–1.2 M discontinuous sucrose gradient and centrifuged at 85,000 g for 120 min at 4 °C in a Beckman SW41-Ti rotor. The synaptic membrane (SM) fraction was collected at the 1.0–1.2 M interface, diluted with 5 mL 25 mM Tris pH7.5, and spun at 78,500g for 20 min at 4 °C in a Beckman SW55-Ti rotor. The resulting pellet was homogenized in 0.8 mL 10 mM Bicine pH 7.5/5% n-octyl-β-D-glucoside (NOG) in a dounce homogenizer, and 1/10 of the volume was retained as the SM fraction. The rest was layered onto a 1–1.4–2.2 M discontinuous sucrose gradient and centrifuged at 85,000g for 120 min at 4 °C in a Beckman SW55-Ti rotor. The PSD fraction was collected from the 1.4–2.2 M interface, diluted with 5 mL 10 mM Bicine, and spun at 78,500g for 20 min at 4 °C in a Beckman SW55-Ti rotor. The pellet containing the PSD was dissolved in 50 μL 5% sodium dodecyl sulfate (SDS) and sonicated for 10 s at 20% amplitude.

### Solubilization of the postsynaptic density and Triton-insoluble proteins by SDS denaturation and subsequent renaturation for downstream immunoprecipitation

Triton-insoluble proteins were denatured and renatured as described[95]. The Triton-insoluble pellet was resuspended in 7 volumes of denaturation buffer containing 50 mM Tris pH7.5, 2% SDS, 1× protease- and 1× phosphatase-inhibitors, and further solubilized by sonication for 10 s at 20% amplitude at room temperature. After centrifugation at 13,000 rpm for 1 h at room temperature, Chaps at a final concentration of 0.1% was added to the supernatant. SDS was precipitated by the addition of SDS-Out SDS Precipitation Reagent (Thermo Fisher Scientific) at a volume equivalent to 1/20 the volume of the supernatant, followed by incubation for 20 min and centrifugation at 10,000 rpm for 10 min, both at 4 °C. The resulting supernatant was cleared of any residual precipitates by passage through a Spin-X column (45 μm pore size; Corning Inc.).

For renaturation of proteins, 1/2 volume of cold renaturation buffer (100 mM Tris pH7.5, 2.4% Tween-20, 1× protease- and 1× phosphatase-inhibitors) was added, and the sample was incubated for 1.5 h rotating at 4 °C before the addition of 1/4 volume of 3.06% cycloamylose (Sigma-Aldrich). The sample was incubated overnight rotating at 4 °C, after which the supernatant containing solubilized proteins was collected by centrifugation, and protein concentration was determined by Bio-Rad DC assay (Bio-Rad Laboratories, Hercules, CA).

### Biochemical detection of protein ubiquitination

In total, 200–600 μg of total renatured proteins were diluted in 2–3 volumes of immunoprecipitation (IP) buffer (1× Tris-buffered saline (TBS), 0.1% Chaps, 1× protease- and 1× phosphatase-inhibitors), combined with respective primary antibodies (Supplementary Table 1), and incubated overnight at 4 °C with gentle agitation. Then, antibody-bound proteins were coupled to Dynabeads Protein G beads (30 μL, Thermo Fisher Scientific) by gentle agitation for 1.5 h at 4 °C. Beads were washed three times with IP buffer and resuspended in 30 μL SDS-sample buffer (62 mM Tris pH6.8, 10% glycerol, 2% SDS, bromophenol blue, 50 mM DTT). Samples combined with isotype control antibodies served as non-specific binding controls.

Samples were loaded on 4–12% Tris-Glycine gels (Thermo Fisher Scientific), and proteins were separated by SDS-PAGE, transferred to PVDF membranes (EMD Millipore, Burlington, MA), and detected by western blotting. For each IP, 10 μg protein was loaded as input control. After transfer, membranes were blocked in 5% (w/v) nonfat milk in 1× TBS, 0.1% Tween-20 (TBST) and incubated overnight with primary antibodies (Supplementary Table 2) in 0.3–5% BSA in TBST at 4 °C. Membranes were washed in TBST, incubated with HRP-linked secondary antibodies diluted in TBST for 1 h at room temperature, and washed again prior to detection of proteins with a ChemiDoc imager (Bio-Rad Laboratories).

### Biochemical detection of protein phosphorylation

For detection of phosphorylated proteins by Western Blotting, Triton-insoluble extracts obtained from cortical tissue of mice that underwent MCAO and 1 h reperfusion were solubilized in 5 volumes 5% SDS and sonicated for 5 s at 20% amplitude at room temperature. Protein concentration was measured with Bio-Rad DC assay. In total, 5–30 μg protein was resolved on 4–12% Tris-Glycine gels, transferred to PVDF membranes, and detected as described above. For detection of phosphorylated Pyk2, Pyk2 protein was immunoprecipitated with anti-Pyk2 antibody from 50 μg Triton-insoluble extracts resolved in RIPA buffer (50 mM Tris pH8, 150 mM NaCl, 1 mM EDTA, 1% Triton X100, 0.1% SDS, 1× protease and 1× phosphatase inhibitors), and phospho-tyrosine was detected by Western Blotting. All used antibodies are listed in Supplementary Tables 1 and 2.

### CaMKII, PKC, Cdk5, CKβ, and Pyk2 kinase activity assays

For all kinase activity measurements, PSD lysates were obtained from 4 mm mouse cortical strips by homogenization in 200 μL ice-cold lysis buffer (50 mM Tris pH7.5, 150 mM NaCl, 25% glycerol, 0.25 mM DTT, 0.1 mM EDTA, 0.1 mM EGTA, 1× protease, and 1× phosphatase inhibitors) using a glass dounce homogenizer. Then 2% Triton X100 and 150 mM KCl were added. Lysates were incubated for 30 min on ice, after which the pellet fractions containing the PSD were obtained by centrifugation for 10 min at 13,000 rpm at 4 °C.

For measuring CaMKII and PKC activities, the PSD pellet was re-homogenized in 150 μL lysis buffer by bouncing 50 times. CaMKII activity was assessed by its ability to transfer a radioactive phosphate from $\gamma^{32}$P-ATP (Perkin Elmer, Shelton, CT) onto the highly selective peptide substrate Autocamtide-2 (amino acid sequence KKALRRQETVDAL; Abbiotec, Escondido, CA) as described[96]. For determining PKC activity, we used a similar approach. Briefly, 10 μg brain PSD-lysate was incubated with 40 μL reaction buffer containing 10 mM HEPES pH7.4, 5 mM MgCl2, 0.5 mM DTT, 1.6 mM CaCl$_2$, 200 μM phosphatidylserine, 16 μM diacylglycerol, 0.03% Triton X100, 40 μM PKC substrate peptide (amino acid sequence ERMRPRKRQGSVRRRV; AnaSpec, Fremont, CA), 50 μM ATP, and 5 Ci/ mmole $\gamma^{32}$P-ATP. To ensure signal specificity, 10 μM PKC inhibitor Gouml 6983 (Abcam, Cambridge, UK) was added to some samples. Reactions were incubated at 30 °C for 1 min, after which they were terminated by spotting on either P81 phosphocellulose cation exchange chromatography paper (Whatman) or phosphocellulose paper manufactured by St. Vincent's Institute (SVI, Fitzroy, Australia). Filters were washed 4 times for 2 min in 0.5% PA, and the remaining radioactivity was quantified in an LS3801 scintillation counter (Beckman Colter) by the Cherenkov method.

Cdk5 activity was measured after immunopurification of Cdk5 with p25/p35 antibody from PSD lysates as reported in[97]. PSD pellets were resuspended in 150 μL RIPA buffer, and Cdk5 was pulled down from 500 μg total protein. Bead-coupled Cdk5 was mixed with 40 μL reaction buffer (50 mM HEPES pH7.4, 5 mM MgCl$_2$, 0.05% BSA, 50 μM histone H1 peptide substrate (amino acid sequence PKTPKKAKKL, Enzo Life Sciences, Farmingdale, NY), 50 μM ATP, 1 mM DTT, and 5 Ci/mmole $\gamma^{32}$P-ATP). Reactions were incubated at 30 °C for 30 min with gentle agitation every 10 min to keep the beads suspended and stopped as described above. The specificity of substrate phosphorylation by Cdk5 was verified by adding 10 μM Cdk5 inhibitor ((R)-CR8, Tocris Bioscience, Bristol, UK) to companion reactions.

CKβ activity was measured with the creatine kinase activity assay kit from Sigma-Aldrich according to the manufacturer's instructions. In short, brains used for CKβ measurements were washed in 1× PBS to remove blood that could contain high levels of CKβ and confound results. After isolation, PSD pellets were resuspended in 100 μL lysis buffer and brought to a concentration of 10 μg/mL. In total, 10 μL samples were combined with 100 μL assay buffer, 10 μL substrate, and 1 μL enzyme mix, all provided by the kit. For blank and calibrator controls, 110 μL ddH$_2$O or 100 μL ddH$_2$O and 10 μL assay calibrator, respectively, were used. Absorbance at 340 nm was measured after 20 (A initial) and 40 min (A final) incubation of reactions at room temperature. CKβ activity in units/L was determined through the following equation: ((A final sample − A initial sample)/(A initial calibrator − A initial blank)) × 150. All measurements were performed in duplicates.

Pyk2 activity was assessed through its autophosphorylation level and capacity to phosphorylate recombinant mouse his-GST-Src (Sino Biological

Inc., Chesterbrook, PA). For autophosphorylation, cortical PSD pellets were obtained from sham brains as well as different time points after MCAO, and PSD proteins were extracted in 150 μL RIPA buffer. 10 μg lysates were loaded on SDS-PAGE and stained for Pyk2 Y402. Phosphorylation levels were calculated relative to total levels of Pyk2. In addition, overall tyrosine phosphorylation of Pyk2 was determined after immunopurification of the kinase and probing with a phospho-tyrosine antibody. For Src substrate phosphorylation, recombinant mouse his-GST-Src was first immobilized on glutathione sepharose 4B beads (GE Healthcare). 20 μg purified protein was resolved in 460 μl 1×PBS and incubated with beads for 2 h at 4 °C with light agitation. Beads were washed, immobilized Src was taken up in 80 μl 1×PBS containing protease inhibitors and stored at 4 °C until successful binding was confirmed by Western Blot analysis. For the kinase assay, Pyk2 was immunopurified from 50 μg PSD brain extracts and incubated with 8 μl GST-Src-beads for 1 h at 30 °C in 300 μl kinase buffer (40 mM Tris pH7.5, 20 mM MgCl$_2$, 0.1 mg/ml BSA, 2.5 mM MnCl$_2$, 2 mM DTT, 10 mM ATP) in presence of protease and phosphatase inhibitors. The reaction was stopped by washing beads in $1 \times$ PBS and resolving them in 1x SDS-sample buffer. Samples were loaded on SDS-PAGE and phosphorylated and total Src were detected by Western Blotting with respective antibodies.

To determine the role of ubiquitin in regulating kinase activities, lysates and precipitates were treated with recombinant deubiquitinase USP2 catalytic domain (Enzo Life Sciences) for 1 h at 37 °C at a ratio of 1: 5 μg USP2: lysate. Removal of high molecular weight conjugated ubiquitin was verified by Western Blotting. For all measurements, kinase activity levels were quantified relative to protein levels in respective lysates and precipitates.

## Statistics and reproducibility

Sham and MCAO surgeries were performed in randomly assigned mice, and data analyses were performed blinded whenever possible. MS, biochemical ubiquitination, and phosphorylation assessments, as well as kinase activity experiments, were performed in separate groups of animals. For biochemical ubiquitination experiments, due to their comparable ubiquitination pattern observed by MS, sham and contralateral samples were used as interchangeable controls.

Comparisons between 2 groups were statistically evaluated by the student's unpaired or paired $t$ test, as appropriate. Comparisons between multiple groups were performed with one-way ANOVA followed by Bonferroni post hoc test. Differences were considered significant at $P < 0.05$. For each performed experiment, the exact sample size, applied statistical test, and the resultant $P$ value is presented in the figure legend. At least three independent experiments were performed to support each finding in the manuscript that was statistically evaluated. For quantification of Pyk2 Y402 phosphorylation, which was variably elevated across reperfusion times, we performed outlier analysis and exclusion by the interquartile range (IQR) method ($\pm 1.5 \times$ IQR).

## Reporting summary

Further information on research design is available in the Nature Portfolio Reporting Summary linked to this article.

## Data availability

The mass spectrometry proteomics data have been deposited to the ProteomeXchange Consortium (http://proteomecentral.proteomexchange.org) via the PRIDE partner repository with the dataset identifier PXD042951, which is publicly available. Proteomics ubiquitination data can also be interrogated via a user-friendly database created on the shinyapps.io platform (https://hochrainerlab.shinyapps.io/StrokeUbiOmics/). Extracted proteomics source data converted to log-scale changes can be found in Supplementary Data 1. Source data underlying analyses for plots and graphs are detailed in Supplementary Data 3. Images of uncropped and unedited Western blots are provided in Supplementary Figs. 8–11. Additional information is available from the corresponding author upon reasonable request.

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

## Acknowledgements

This study was supported by National Institute of Health (NIH) grants R01-NS109588 to K.H., R01-NS34179 to C.I. and SIG 1S10 OD017992-01 to S.Z., as well as a Weill Cornell Sackler Brain and Spine Institute grant to K.H., and a post-doctoral research grant from the Deutsche Forschungsgemeinschaft (KA3810/1-1) to A.K. Support from the Feil Family Foundation is gratefully acknowledged. The authors wish to thank Christina DiMauro, Ashley Roth, and Ashley Hansen (WCM, New York, NY) for technical help and Qin Fu (Institute of Biotechnology, Cornell University, Ithaca, NY) for help with the PRIDE upload.

## Author contributions

C.P., E.H., A.K. and C.B. performed MCAO surgeries; K.H., V.P., E.H., L.D., I.B., V.B. and A.H. conducted molecular experiments and associated analyses; S.Z. and R.B. performed nanoLC–MS/MS; K.H., U.R. and R.B. analyzed MS data; L.D. created the searchable web interface; C.I. and K.H. provided funding; K.H. conceived and supervised the research, and wrote the paper.

## Competing interests

The authors declare no competing interests.
