## [Peer review file · Communications Biology]

Reviewers' comments:

Reviewer #1 (Remarks to the Author):

In this manuscript, Dhawka et al. examined global changes of the ubiquitin landscape in response to ischemic stroke. Briefly, the authors performed ubiquitin remnant (diGly) profiling of detergent (Triton)-resistant neocortex fractions from mice undergoing sham surgery or transient middle cerebral artery occlusion (tMCAO) followed by reperfusion for 1 h. By comparing ipsi- and contralateral (and sham) cortices, the authors detected substantially increased ubiquitination sites in the former while total proteome abundance did not change. Interestingly, proteins expressed in neurons and found at the postsynaptic density (PSD) featured most prominently among the 208 differentially ubiquitinated proteins. The authors confirmed increased ubiquitination of a number of candidates including receptors (GluN1), scaffolding (PSD95) and adaptor (Shank2) proteins as well as kinases (CaMKII, PKC, Cdk5, Pyk2) using IP and immunoblotting of cortical PSD fractions. To probe the potential ubiquitin-mediated regulation of kinases upon ischemic stroke, the authors performed phospho-proteomics on the same cortical fractions. Consistent with their hypothesis, the authors identified increase (tyrosine) phosphorylation pattern at the PSD. The authors went on to validate a number of stroke-induced phosphorylation events of established CaMKII, PKC, Cdk5 and Pyk2 substrates using phosphosite specific antibodies. Lastly, the authors showed that some of these events can be reversed by treating homogenates of cortical fraction with a promiscuous deubiquitinase. Overall, this elegant work from the Hochrainer lab provides convincing evidence that ubiquitination plays a role in post-ischemic kinase regulation at the PSD. Since this is a fairly comprehensive study, I only have a few critical points to raise.

- 1) Given the low number (369) of ubiquitinated proteins in the detergent-resistant cortices fractions, it would make a more complete picture if the authors extend their ubiquitination site profiling beyond these fractions and monitor the whole cortex. This is in particular important since the authors kind of pre-select the subcellular compartment by focusing on this fraction. Also, Figure 1B seems to show the ubiquitin response of the "entire cerebral cortex" and not only of the detergent-resistant fractions. Along similar lines: Have the IPs shown in Figure 3D and 4B be done from detergent-resistant fractions? If this was the case, it should be clearly stated in the figure legend.
- 2) For the phospho-proteome a numeric analysis is missing. How many phospho sites (and which kind) were detected? How many were up and down regulated?
- 3) For all GO term analyses, it would be important to add the information on how many proteins were in each of the enriched categories. This can easily be incorporated by switching to a Dot blot representation (which allows to display to components using color and size).
- 4) Do the authors find any ubiquitin system components (e.g. E3 ligases) differentially phosphorylated?

Reviewer #2 (Remarks to the Author):

Dhawka and Palfini et al. investigate the post-translational modifications, especially proteins involved in the postsynaptic density, following whole-brain ischemia using a proteomic approach. First, they identify the post-ischemic ubiquitin proteome by ubiquitin enrichment combined with LC-MS/MS and find out most of the proteins with increased ubiquitination levels are focused in the postsynaptic density (PSD). Next, they show that ubiquitination regulates the phosphorylation and subsequently the activity of the PSD kinases. The proteins identified in this study could be potential targets for therapeutic strategies used to treat ischemic brains. Please find my main comments below:

1. My biggest concern with this paper is the general use of "ubiquitination" and "ubiquitinated proteins," while "Ubiquitination" is a very broad term. There are many different types of ubiquitination, and each type has a different function in the cell. Even though some studies use the

non-biased method to identify proteins modified by any type of ubiquitin chains, it does not provide much useful information since each type has a totally different signal for the cell; it could be a signal for protein degradation, or for DNA damage, or NFkB transcription regulation, etc. I understand this paper still carries interesting and important findings, but it leaves many questions unanswered and open.

2. Even if they used a general approach to study the ubiquitination, it'd help the manuscript very much if the authors tried to at least identify the type of ubiquitination for the major proteins they found interesting and focused on them toward the end of the work (like the PSD kinases or NMDA receptor subunits). This way, it would reveal how each protein being activated/inactivated is being controlled by a type of ubiquitination.

3. No statistics are provided for many of the figures, like Figure 1 B (western blotting). The authors should provide the statistics of each experiment (not selectively) in the text or in the figure's caption.

4. Page 16, line 9: The sentence "Confirming this prediction, ..." does not seem to have a verb.

5. Figure 3C: There is an extra legend at the bottom left (Ub increased)

6. The experimental designs are not explained well in the result section (there are some explanations in the figure captions). For example, for Figure 3, it's not clear if they used the same tissues as Figure 1 or if they used new animals and groups following the same protocol. The rationalization for doing each experiment is also excluded, forcing the reader to guess the reason behind each experiment. Finally, they excluded the contralateral group from these analyses without providing any explanations.

7. The result sections lack summary/conclusion sentences explaining what each experiment/data means.

8. In the volcano plots, the contra- is compared with the sham group, but the ipsi- is compared with sham + contra. Why is that?

9. Page 23, line 14-15: The authors cite four papers to show that "PSD95 ubiquitination regulates AMPA receptor internalization, synapse elimination, PSD assembly and plasticity." While it's true, all the references are about proteasome-dependent ubiquitination (K48-polyubiquitin). So, basically, it's the PSD95 degradation that matters, not its ubiquitination, which makes sense! PSD95 is needed for the normal function of the PSD and the synapse so its absence will change the synaptic functions. So, the authors should be careful when interpreting their data.

10. Page 23, line 30-32: the lack of reduced protein detection levels (by western blotting) does not necessarily indicate the ubiquitination is non-degradative. It is possible that the proteins will take some time to be degraded (longer than 1 hour, like 4 hours). Unless the authors can provide evidence from other studies showing the degradation rate of these specific proteins is around 1 hour, additional experiments are needed to show the protein levels did not decrease even after 2-4 hours.

Reviewer #3 (Remarks to the Author):

In this manuscript from Dhawka et al., the research follows on from the lab's previous work in the field. Novelty in the current manuscript is afforded through the identification of global ubiquitome changes in the MCAO model of ischemia and mechanistically through understanding of how alterations in specific ubiquitinated proteins modify the activity of target proteins in the post synaptic density.

Of benefit to the scientific community, the authors have generated a searchable database to identify MCAO-driven ubiquitinated proteins of interest and the methods are extremely well described. Ischemic stroke led to increased ubiquitination of 198 proteins with highest expression levels in glutamatergic neurons and identified a large number of ubiquitinated proteins at the post synaptic density. A subset, for example NMDAR, AMPAR and PSD kinases, were validated by IP with ubiquitin.

Following on from this, the authors demonstrated that the increased ubiquitination led to alterations in phosphotyrosine modification of a number of kinases at the PSD, including CDK5, PKC and CamKII, and subsequent reductions in their activity in the ipsilateral MCAO PSD. The addition of the deubiquitinase USP2 reduced the phosphotyrosine signal and increased the activity

of the kinases, demonstrating the reversal of the phenotype.

This manuscript is well written, results are clearly presented and described and will benefit the research community.

Amendments required:

Amendments to the statistical testing and/or descriptions are required.

Fig 5E comparisons were made using t-test but described significant differences from s and c. Test used should have been one way ANOVA as there are three groups for comparison. Please describe what # denotes.

As above, Fig 6A t-test from sham but three groups compared, similarly Fig 7A, 8A-D. Furthermore for 8B-D it would be beneficial to compare the data from 60 min to 30 min, hence one-way ANOVA multiple comparisons test is required.

(Compare to Methods, p.12 line 33: Comparisons between 2 groups were statistically evaluated by the student's unpaired or paired t test, as appropriate. Comparisons between multiple groups were performed with 1-way ANOVA followed by Bonferroni post hoc test.)

We are grateful to the reviewers for their overall positive assessment of our manuscript, and wish to thank them for their thoughtful comments and suggestions for improvement. A detailed response to questions is provided below. Relevant sections in the manuscript are highlighted in red.

Reviewer #1 (Remarks to the Author):

In this manuscript, Dhawka et al. examined global changes of the ubiquitin landscape in response to ischemic stroke. Briefly, the authors performed ubiquitin remnant (diGly) profiling of detergent (Triton)-resistant neocortex fractions from mice undergoing sham surgery or transient middle cerebral artery occlusion (tMCAO) followed by reperfusion for 1 h. By comparing ipsi- and contralateral (and sham) cortices, the authors detected substantially increased ubiquitination sites in the former while total proteome abundance did not change. Interestingly, proteins expressed in neurons and found at the postsynaptic density (PSD) featured most prominently among the 208 differentially ubiquitinated proteins. The authors confirmed increased ubiquitination of a number of candidates including receptors (GluN1), scaffolding (PSD95) and adaptor (Shank2) proteins as well as kinases (CaMKII, PKC, Cdk5, Pyk2) using IP and immunoblotting of cortical PSD fractions. To probe the potential ubiquitin-mediated regulation of kinases upon ischemic stroke, the authors performed phospho-proteomics on the same cortical fractions. Consistent with their hypothesis, the authors identified increase (tyrosine) phosphorylation pattern at the PSD. The authors went on to validate a number of stroke-induced phosphorylation events of established CaMKII, PKC, Cdk5 and Pyk2 substrates using phosphosite specific antibodies. Lastly, the authors showed that some of these events can be reversed by treating homogenates of cortical fraction with a promiscuous deubiquitinase. Overall, this elegant work from the Hochrainer lab provides convincing evidence that ubiquitination plays a role in post-ischemic kinase regulation at the PSD. Since this is a fairly comprehensive study, I only have a few critical points to raise.

Q1: Given the low number (369) of ubiquitinated proteins in the detergent-resistant cortices fractions, it would make a more complete picture if the authors extend their ubiquitination site profiling beyond these fractions and monitor the whole cortex. This is in particular important since the authors kind of pre-select the subcellular compartment by focusing on this fraction. Also, Figure 1B seems to show the ubiquitin response of the “entire cerebral cortex” and not only of the detergent-resistant fractions. Along similar lines: Have the IPs shown in Figure 3D and 4B be done from detergent-resistant fractions? If this was the case, it should be clearly stated in the figure legend.

Response: We apologize to the reviewer if the rationale for pre-selection of detergent-insoluble fractions for post-stroke ubiquitin proteomics was not made sufficiently clear. The decision to focus on detergent-insoluble fractions was based on previous findings that ischemia with reperfusion leads to upregulated ubiquitination only in detergent-resistant fractions, while it is completely absent in soluble fractions¹⁻³. In addition, with this knowledge in mind, we found that restricting our approach to the detergent-insoluble portion rather than the entire tissue lysate reduced sample complexity, which is a major limitation in detecting post-translational modifications, such as ubiquitination, by proteomics^{4,5}. Regarding the number of identified proteins, in 3 mass spectrometry runs we detected 1047 unique protein hits, which is on par with other rodent brain tissue ubiquitome studies⁶⁻⁹. To only consider the highest confidence hits, we restricted our post analysis to the 369 proteins that were called in all 3 runs. As with proteomics, the Western Blot and IPs underlying Figures 1B, 3D, and 4B, were performed with the detergent-insoluble cortical fraction, and we have now labelled this accordingly in the figure legends to Figures 1, 3, 4, and 5 (pages 14, 17, 18, and 19). We have also added statements detailing the sample selection (methods section page 8, lines 24-29; and results section page 13, lines 18-20 and 32-33; and page 17, lines 1 and 15).

Q2: For the phospho-proteome a numeric analysis is missing. How many phospho sites (and which kind) were detected? How many were up and down regulated?

Response: We detected a total 857 phospho-sites in the detergent-insoluble fraction, 8 of which were down- and 66 upregulated (2-fold change, $P < 0.05$). We have included this information in Figure 5 and the corresponding figure legend (page 19). From the sites that could be clearly identified, 612 were serine, 38 were threonine, and 8 were tyrosine sites. This information can be found in Table S1.

Q3: For all GO term analyses, it would be important to add the information on how many proteins were in each of the enriched categories. This can easily be incorporated by switching to a Dot blot representation (which allows to display to components using color and size).

Response: Thank you for this suggestion. We now are showing the GO analysis in Dot blot format including the requested information, as well as the enrichment score. Please see Figures S2 and 5.

Q4: Do the authors find any ubiquitin system components (e.g., E3 ligases) differentially phosphorylated?

Response: This is an interesting question, since many ubiquitination enzymes are regulated by phosphorylation. We now interrogated our dataset (shown in Table S1) for such changes. We detected phosphorylation sites in five E3 ligases, including HectD1, Hecw2, Mib2, Nedd4 and Ufl1, and one deubiquitinase, Vcpi1. After applying our selection criteria (2-fold change, $P < 0.05$), Nedd4 and Vcpi1 exhibited altered phosphorylation status after ischemia (Nedd4 T287, upregulated; and Vcpi1 unknown site within amino acids 987-1004 (AEATTRSRESSPSHGLLK), upregulated). Whether these changes also lead to altered activity of these enzymes is a great suggestion for future investigations. We have mentioned this point in the discussion (page 25, lines 3-6).

Reviewer #2 (Remarks to the Author):

Dhawka and Palfini et al. investigate the post-translational modifications, especially proteins involved in the postsynaptic density, following whole-brain ischemia using a proteomic approach. First, they identify the post-ischemic ubiquitin proteome by ubiquitin enrichment combined with LC-MS/MS and find out most of the proteins with increased ubiquitination levels are focused in the postsynaptic density (PSD). Next, they show that ubiquitination regulates the phosphorylation and subsequently the activity of the PSD kinases. The proteins identified in this study could be potential targets for therapeutic strategies used to treat ischemic brains. Please find my main comments below:

Q1: My biggest concern with this paper is the general use of “ubiquitination” and “ubiquitinated proteins,” while “Ubiquitination” is a very broad term. There are many different types of ubiquitination, and each type has a different function in the cell. Even though some studies use the non-biased method to identify proteins modified by any type of ubiquitin chains, it does not provide much useful information since each type has a totally different signal for the cell; it could be a signal for protein degradation, or for DNA damage, or NFkB transcription regulation, etc. I understand this paper still carries interesting and important findings, but it leaves many questions unanswered and open.

Response: We thank the reviewer for this comment and agree about the importance of ubiquitin chain type assembly for functional consequences of ubiquitination. However, the current study was designed to provide an initial large-scale assessment of the ubiquitome after ischemic stroke,

and not to identify ubiquitin chain types conjugated to specific proteins. This is challenging to accomplish by proteomics and would require a very specialized approach, such as combination with ubiquitin-site mutants or pulldown with chain-specific antibodies, that to our knowledge has never been carried out with complex tissue samples. However, the point raised by the reviewer is well taken and to gain some insight into changes in ubiquitin chain composition associated with stroke, we now determined K6, K11, K27, K29, K33, K48 and K63-modified ubiquitin peptide abundances in control versus ipsilateral stroke samples. We found that while K27, K29, and K33-chains were only minimally affected, K6, K11, K48 and K63-chains were highly enriched after stroke. This confirms earlier data from our lab showing upregulation of both K48- and K63-linked ubiquitin by Western Blotting with chain-specific antibodies², and provides further evidence for changes in multiple ubiquitin-associated pathways after ischemic stroke. We included these novel data as Figure S1 and description in the manuscript text (page 14, lines 1-6).

Q2: Even if they used a general approach to study the ubiquitination, it'd help the manuscript very much if the authors tried to at least identify the type of ubiquitination for the major proteins they found interesting and focused on them toward the end of the work (like the PSD kinases or NMDA receptor subunits). This way, it would reveal how each protein being activated/inactivated is being controlled by a type of ubiquitination.

Response: Thank you for this suggestion. We now performed immunoprecipitation of PSD kinases followed by staining with K48- and K63-ubiquitin linkage-specific antibodies to explore potential degradative and non-degradative functions of ubiquitin attachment to kinases after stroke. Interestingly, we observed very different scenarios for each inspected kinase. While CaMKII α was modified with both ubiquitin chain types, PKC β was conjugated to neither. Although there was a tendency of Pyk2 and Cdk5 modification with K63-ubiquitin, these kinases were preferentially attached to K48-ubiquitin. We included these new data as Fig. S5 and described the findings in the manuscript text (page 17, lines 16-20; page 24, lines 26-28).

Q3: No statistics are provided for many of the figures, like Figure 1B (western blotting). The authors should provide the statistics of each experiment (not selectively) in the text or in the figure's caption.

Response: We now included quantifications for the Western Blot in Figure 1B, and the immunoprecipitation experiments in Figures 3D and 4B. We also statistically evaluated the data, where indicated. The Western Blot in Figure 1B only served as proof-of-principle showing that ubiquitination was elevated across the cortical area used for the mass-spectrometry experiment. It was not intended for statistical evaluation. A statistically significant increase of cortical ubiquitination in this ischemia model has been extensively shown and is well established in the literature^{2,10,11}. We have updated Figures 1 (page 14), 3 (page 16), and 4 (page 18) as well as corresponding figure legends with the requested information.

Q4: Page 16, line 9: The sentence "Confirming this prediction, ..." does not seem to have a verb.

Response: We corrected the sentence (now page 15, line 11).

Q5: Figure 3C: There is an extra legend at the bottom left (Ub increased)

Response: Thank you. We removed this piece of legend and corrected the figure (page 16).

Q6: The experimental designs are not explained well in the result section (there are some explanations in the figure captions). For example, for Figure 3, it's not clear if they used the same tissues as Figure 1 or if they used new animals and groups following the same protocol. The rationalization for doing each experiment is also excluded, forcing the reader to guess the reason behind each experiment. Finally, they excluded the contralateral group from these analyses without providing any explanations.

Response: We apologize if the underlying experimental design and rationale was not made clear in each results section. We used different groups of mice for proteomics in Figure 1 and each experiment in Figure 3. Since we found ubiquitination to be similar in sham and contralateral groups (Figure 2A; see also response to Q8 below), we used either group as control for biochemical experiments investigating ubiquitination in Figures 3 and 4. We now amended the manuscript to include this and other relevant information about design and rationale of chosen experiments. Please see methods section (page 4, lines 15-18) and results section (page 13, lines 31-32 and 35-36; page 15, lines 7-8; page 17, lines 6-7; page 20, line 18; page 22, line 7).

Q7: The result sections lack summary/conclusion sentences explaining what each experiment/data means.

Response: We now added a summary statement at the end of each results section (page 15, lines 3-4; page 17, lines 2-3; page 20, lines 3-4; page 22, lines 3-4 and line 17).

Q8: In the volcano plots, the contra- is compared with the sham group, but the ipsi- is compared with sham + contra. Why is that?

Response: Focal ischemic stroke can cause changes in the contralateral side, which sometimes precludes its use as comparative control to the ipsilateral side. To evaluate whether this is the case in our study, we asked whether the sham and contralateral groups exhibit large differences in ubiquitination (Figure 2A, left volcano plot). Since this was not the case, we used an average control value derived from sham + contralateral samples to evaluate changes in the ipsilateral group (Figure 2A, right volcano plot). We added a sentence in the methods section to explain this rationale (page 9, lines 1-2).

Q9: Page 23, line 14-15: The authors cite four papers to show that “PSD95 ubiquitination regulates AMPA receptor internalization, synapse elimination, PSD assembly and plasticity.” While it’s true, all the references are about proteasome-dependent ubiquitination (K48-polyubiquitin). So, basically, it’s the PSD95 degradation that matters, not its ubiquitination, which makes sense! PSD95 is needed for the normal function of the PSD and the synapse so its absence will change the synaptic functions. So, the authors should be careful when interpreting their data.

Response: We amended the section to acknowledge the proteasome-dependent (references 69-71) and independent (reference 58) role of PSD95 ubiquitination on PSD function (now page 24, line 10).

Q10: Page 23, line 30-32: the lack of reduced protein detection levels (by western blotting) does not necessarily indicate the ubiquitination is non-degradative. It is possible that the proteins will take some time to be degraded (longer than 1 hour, like 4 hours). Unless the authors can provide evidence from other studies showing the degradation rate of these specific proteins is around 1hour, additional experiments are needed to show the protein levels did not decrease even after 2-4 hours.

Response: We concur with the reviewer and removed this statement from the discussion section.

Reviewer #3 (Remarks to the Author):

In this manuscript from Dhawka et al., the research follows on from the lab’s previous work in the field. Novelty in the current manuscript is afforded through the identification of global ubiquitome changes in the MCAO model of ischemia and mechanistically through understanding of how alterations in specific ubiquitinated proteins modify the activity of target proteins in the post synaptic density.

Of benefit to the scientific community, the authors have generated a searchable database to identify MCAO-driven ubiquitinated proteins of interest and the methods are extremely well described. Ischemic stroke led to increased ubiquitination of 198 proteins with highest expression levels in glutamatergic neurons and identified a large number of ubiquitinated proteins at the post synaptic density. A subset, for example NMDAR, AMPAR and PSD kinases, were validated by IP with ubiquitin.

Following on from this, the authors demonstrated that the increased ubiquitination led to alterations in phosphotyrosine modification of a number of kinases at the PSD, including CDK5, PKC and CamKII, and subsequent reductions in their activity in the ipsilateral MCAO PSD. The addition of the deubiquitinase USP2 reduced the phosphotyrosine signal and increased the activity of the kinases, demonstrating the reversal of the phenotype.

This manuscript is well written, results are clearly presented and described and will benefit the research community.

Amendments required:

Amendments to the statistical testing and/or descriptions are required.

Q1: Fig 5E comparisons were made using t-test but described significant differences from s and c. Test used should have been one way ANOVA as there are three groups for comparison. Please describe what # denotes.

Response: Thank you for spotting this mistake. We now performed one-way ANOVA analysis for all comparisons in Figure 5 (page 19). In addition, we removed '#' from the figure.

Q2: As above, Fig 6A t-test from sham but three groups compared, similarly Fig 7A, 8A-D. Furthermore, for 8B-D it would be beneficial to compare the data from 60 min to 30 min, hence one-way ANOVA multiple comparisons test is required. (Compare to Methods, p.12 line 33: Comparisons between 2 groups were statistically evaluated by the student's unpaired or paired t test, as appropriate. Comparisons between multiple groups were performed with 1-way ANOVA followed by Bonferroni post hoc test.)

Response: Thank you for pointing this out. We now performed one-way ANOVA analyses for these figures (Figure 6, page 20; Figure 7 and 8, page 21).

- 1 Hayashi, T., Takada, K. & Matsuda, M. Subcellular distribution of ubiquitin-protein conjugates in the hippocampus following transient ischemia. *J Neurosci Res* **31**, 561-564 (1992),
- 2 Hochrainer, K., Jackman, K., Benakis, C., Anrather, J. & Iadecola, C. SUMO2/3 is associated with ubiquitinated protein aggregates in the mouse neocortex after middle cerebral artery occlusion. *J Cereb Blood Flow Metab* **35**, 1-5 (2015), DOI: 10.1038/jcbfm.2014.180
- 3 Liu, C. L., Martone, M. E. & Hu, B. R. Protein ubiquitination in postsynaptic densities after transient cerebral ischemia. *J Cereb Blood Flow Metab* **24**, 1219-1225 (2004),
- 4 Chandramouli, K. & Qian, P. Y. Proteomics: challenges, techniques and possibilities to overcome biological sample complexity. *Hum Genomics Proteomics* **2009** (2009), DOI: 10.4061/2009/239204
- 5 Dupree, E. J. et al. A Critical Review of Bottom-Up Proteomics: The Good, the Bad, and the Future of this Field. *Proteomes* **8** (2020), DOI: 10.3390/proteomes8030014
- 6 Na, C. H. et al. Synaptic protein ubiquitination in rat brain revealed by antibody-based ubiquitome analysis. *J Proteome Res* **11**, 4722-4732 (2012), DOI: 10.1021/pr300536k
- 7 Iwabuchi, M. et al. Characterization of the ubiquitin-modified proteome regulated by transient forebrain ischemia. *J Cereb Blood Flow Metab* **34**, 425-432 (2014), DOI: 10.1038/jcbfm.2013.210

- 8 Cai, Y. et al. Comprehensive analysis of the ubiquitome in rabies virus-infected brain tissue of *Mus musculus*. *Vet Microbiol* **241**, 108552 (2020), DOI: 10.1016/j.vetmic.2019.108552
- 9 He, F. et al. Global ubiquitome analysis of substantia nigra in doubly-mutant human alpha-synuclein transgenic mice. *Behav Brain Res* **380**, 112436 (2020), DOI: 10.1016/j.bbr.2019.112436
- 10 Hochrainer, K., Jackman, K., Anrather, J. & Iadecola, C. Reperfusion rather than ischemia drives the formation of ubiquitin aggregates after middle cerebral artery occlusion. *Stroke* **43**, 2229-2235 (2012), DOI: 10.1161/STROKEAHA.112.650416
- 11 Hu, B. R. et al. Protein aggregation after focal brain ischemia and reperfusion. *J Cereb Blood Flow Metab* **21**, 865-875 (2001),

REVIEWERS' COMMENTS:

Reviewer #1 (Remarks to the Author):

The authors have addressed all of my concerns. They have significantly improved the manuscript and provided more information.

Reviewer #3 (Remarks to the Author):

The authors have addressed all of my comments.